



# The semiannual oscillation (SAO) in the tropical middle atmosphere and its gravity wave driving in reanalyses and satellite observations

Manfred Ern[1], Mohamadou Diallo[1], Peter Preusse[1], Martin G. Mlynczak[2], Michael J. Schwartz[3], Qian Wu[4], and Martin Riese[1]

[1]Institut für Energie- und Klimaforschung – Stratosphäre (IEK–7), Forschungszentrum Jülich GmbH, 52425 Jülich, Germany
[2]NASA Langley Research Center, Hampton, Virginia, USA.
[3]Jet Propulsion Laboratory, California Institute of Technology, Pasadena, California, USA.
[4]National Center for Atmospheric Research, High Altitude Observatory, Boulder, Colorado, USA.

**Correspondence:** M. Ern (m.ern@fz-juelich.de)

**Abstract.** Gravity waves play a significant role in driving the semiannual oscillation (SAO) of the zonal wind in the tropics. However, detailed knowledge of this forcing is missing, and direct estimates from global observations of gravity waves are sparse. For the period 2002–2018, we investigate the SAO in four different reanalyses: ERA-Interim, JRA-55, ERA-5, and MERRA-2. Comparison with the SPARC zonal wind climatology and quasi-geostrophic winds derived from Microwave Limb Sounder (MLS) and Sounding of the Atmosphere using Broadband Emission Radiometry (SABER) satellite observations show that the reanalyses reproduce some basic features of the SAO. However, there are also large differences, depending on the model setup. Particularly, MERRA-2 seems to benefit from dedicated tuning of the gravity wave drag parameterization and assimilation of MLS observations. To study the interaction of gravity waves with the background wind, absolute values of gravity wave momentum fluxes and drag derived from SABER satellite observations are compared with different wind data sets: the SPARC wind climatology, data sets combining ERA-Interim at low altitudes and MLS or SABER quasi-geostrophic winds at high altitudes, as well as data sets that combine ERA-Interim, SABER quasi-geostrophic winds, and direct wind observations by the TIMED Doppler Interferometer (TIDI). In the lower and middle mesosphere SABER absolute gravity wave drag correlates well with positive vertical gradients of the background wind, indicating that gravity waves contribute mainly to the driving of the SAO eastward wind phases and their downward propagation with time. At altitudes 75–85 km, SABER absolute gravity wave drag correlates better with absolute values of the background wind, suggesting a more direct forcing of the SAO winds by gravity wave amplitude saturation. Above about 80 km SABER gravity wave drag is mainly governed by tides rather than by the SAO. The reanalyses reproduce some basic features of the SAO gravity wave driving: All reanalyses show stronger gravity wave driving of the SAO eastward phase in the stratopause region. For the higher-top models ERA-5 and MERRA-2 this is also the case in the lower mesosphere. However, all reanalyses are limited by model-inherent damping in the upper model levels, leading to unrealistic features near the model top. Our analysis of the SABER and reanalysis gravity wave drag suggests that the magnitude of SAO gravity wave forcing is often too weak in the free-running general circulation models, therefore, a more realistic representation is needed.



## 1 Introduction

In the tropics, the zonal wind in the middle atmosphere exhibits characteristic oscillations of semiannual and quasi-biennial
periods. The quasi-biennial oscillation (QBO) has an approximate period of $28\,\mathrm{months}$ and is the dominant mode in the
stratosphere. The semiannual oscillation (SAO) dominates in the upper stratosphere and in the mesosphere with one amplitude
peak in the stratopause region, the stratopause semiannual oscillation (SSAO), and another amplitude peak somewhat below
the mesopause, the mesopause semiannual oscillation (MSAO). For further details regarding the QBO and the SAO please see
Baldwin et al. (2001).

First observations of the SAO winds were made by rocketsondes and radars at single stations in the tropics (e.g., Reed,
1966; Groves, 1972; Hirota, 1978; Dunkerton, 1982; Hamilton, 1982; Palo and Avery, 1993), and observations at tropi-
cal stations are still continued (e.g., Gurubaran and Rajaram, 2001; Venkateswara Rao et al., 2012; Day and Mitchell, 2013;
Kishore Kumar et al., 2014). Direct observations of the SAO winds from satellite were made, for example, by the High Res-
olution Doppler Imager (HRDI) onboard the Upper Atmosphere Research Satellite (UARS) (e.g., Lieberman et al., 1993;
Burrage et al., 1996), or by the Superconducting Submillimeter-Wave Limb-Emission Sounder (SMILES) instrument onboard
the International Space Station (e.g., Baron et al., 2013).

Based on multiple observations including HRDI zonal winds, a first comprehensive climatology of the SAO in the tropical
middle atmosphere was introduced by Garcia et al. (1997). A later assessment led to the Stratospheric Processes And their
Role in Climate (SPARC) global monthly climatology of zonal mean winds (Swinbank and Ortland, 2003; Randel et al., 2002,
2004). Unfortunately, direct global wind observations from satellite in the stratosphere and mesosphere are sparse. Therefore,
Smith et al. (2017) recently investigated whether it is possible to interpolate quasi-geostrophic winds derived from Sounding of
the Atmosphere using Broadband Emission Radiometry (SABER) and Microwave Limb Sounder (MLS) satellite observations
into the tropics. Useful results were obtained for altitudes below about $80\,\mathrm{km}$.

The SAO plays an important role in the whole atmosphere system. Effects of the SAO are also observed in temperatures
(e.g., Reed, 1962; Delisi and Dunkerton, 1988a; Garcia and Clancy, 1990; Huang et al., 2008), and the SAO modulates the
distribution of trace species in the stratosphere (e.g., Shu et al., 2013), as well as in the mesosphere and lower thermosphere
(MLT) (e.g., Huang et al., 2008; Kumar et al., 2011; Zhu et al., 2015). It was found that the QBO and the SAO interact with
each other. For example, the phases of the QBO and SAO can synchronize (e.g., Dunkerton and Delisi, 1997; Krismer et al.,
2013), and eastward phases of the SAO can initiate QBO eastward phases (e.g., Kuai et al., 2009). This effect of the SAO
is of relevance because the QBO couples to the extratropics (e.g., Holton and Tan, 1980; Anstey and Shepherd, 2014), and
has effects on surface weather and climate (e.g., Ebdon, 1975; Marshall and Scaife, 2009; Kidston et al., 2015). Climate and
weather models have difficulties to simulate this influence of the QBO (e.g., Scaife et al., 2014). Further, there is evidence that
both the QBO and the SAO influence the timing of sudden stratospheric warmings (e.g., Pascoe et al., 2006), and a correct
representation of the SAO is needed to explain and better predict such extreme polar vortex events and their influence on
surface weather conditions (Gray et al., 2020). For these reasons it is very important to learn more about the mechanisms that
drive the SAO.



It is known that atmospheric gravity waves contribute to the driving of both the QBO and the SAO. As was shown by several model studies, particularly gravity waves generated by deep convection in the tropics should contribute significantly to the driving of the QBO and the stratopause SAO (e.g., Beres et al., 2005; Kim et al., 2013; Kang et al., 2018), as well as

to the mesopause SAO (e.g., Beres et al., 2005). While critical level filtering of gravity waves of either eastward or westward directed phase speed plays a major role for the driving of the QBO (e.g., Lindzen and Holton, 1968; Lindzen, 1987; Dunkerton, 1997; Baldwin et al., 2001; Ern et al., 2014), the situation is more complicated for the SAO. It was suggested that the forcing of the stratopause SAO should be asymmetric because gravity waves are selectively filtered by the QBO in the stratosphere before entering the altitude range dominated by the SAO (e.g., Hamilton and Mahlmann, 1988; Dunkerton and Delisi, 1997).

The QBO westward phase has a stronger magnitude, and therefore a larger part of the gravity wave spectrum at westward directed phase speeds is filtered out by encountering critical levels. For the stratopause region, this means that the gravity wave spectrum is dominated by eastward propagating waves. Due to this excess of eastward momentum, gravity waves should mainly contribute to the driving of the SAO eastward phase, and only to a lesser extent to the driving of the SAO westward phase. Instead, the driving of the SAO westward phase should be dominated by horizontal advection and the influence of planetary

waves from the extratropics (e.g., Delisi and Dunkerton, 1988b; Hamilton and Mahlmann, 1988).

For the stratopause SAO, this asymmetry was confirmed by High Resolution Dynamics Limb Sounder (HIRDLS) satellite observations of gravity waves (Ern et al., 2015). Semiannual modulations of the global distribution of gravity waves are indeed observed over a large altitude range in the tropical mesosphere (e.g., Kovalam et al., 2006; Krebsbach and Preusse, 2007; Sridharan and Sathishkumar, 2008; Venkateswara Rao et al., 2012; Matsumoto et al., 2016; Chen et al., 2019). However, there

is large uncertainty in which way those gravity waves contribute to the driving of the SAO, and how far the aforementioned asymmetry of gravity wave driving extends upward into the mesosphere. Recent work by Smith et al. (2020) revealed that current global climate models have difficulties in simulating a realistic SSAO. One of the main resaons that was identified is a general lack of eastward forcing by waves in the model — either by large-scale waves, or by gravity waves. Therefore validation of the SAO wave forcing would be required. Another recent study shows that also in current meteorological reanalyses the

SSAO differs strongly between the different reanalyses (Kawatani et al., 2020).

The mesopause SAO is out-of-phase, or even in anti-phase, with the SAO at lower altitudes (e.g., Hirota, 1980; Dunkerton, 1982; Hamilton, 1982). Of course, not only gravity waves, but also advection and medium-scale and global-scale waves (including tides) contribute to the driving of the SAO in the MLT region (e.g., Sassi and Garcia, 1997; Richter and Garcia, 2006). However, likely reason for this out-of-phase relationship is the selective wave filtering of gravity waves by the SSAO and the

SAO in the middle mesosphere. After the selective filtering of the gravity wave spectrum by the background winds, the spectrum is dominated by gravity waves propagating opposite to the wind direction, either eastward or westward, in the middle and lower mesosphere. This is confirmed, for example, by radar observations of gravity wave momentum fluxes (e.g., Matsumoto et al., 2016). If these remaining waves saturate and break in the upper mesosphere and the mesopause region, this results in driving of either the eastward or westward SAO phase, opposite to the wind in the middle mesosphere (e.g., Dunkerton, 1982;

Mengel et al., 1995). This mechanism is also supported by HRDI wind observations (Burrage et al., 1996), as well as by model simulations (see, for example, Richter and Garcia, 2006; Peña–Ortiz et al., 2010). To some extent, even selective wave filter-





ing by the QBO in the stratosphere has effects on the mesopause SAO (e.g., Garcia and Sassi, 1999; Peña–Ortiz et al., 2010). Overall, the driving of the MSAO is not fully understood, and observations of gravity wave momentum flux at the equator are needed to resolve this issue, as stated in a recent review by Vincent (2015).

Our study investigates the SAO and its gravity wave driving in the whole middle atmosphere in the altitude range 30–90 km. We focus on the latitude range 10°S–10°N, and the years 2002–2018 for which satellite data are available. For four reanalyses, the ERA-Interim and ERA-5 reanalyses of the European Centre for Medium-Range Weather Forecasts (ECMWF), the Japanese 55-year Reanalysis (JRA-55) of the Japanese Meteorological Agency (JMA), and the Modern-Era Retrospective Analysis for Research and Applications, Version 2 (MERRA-2) reanalysis of the National Aeronautics and Space Agency

(NASA), we determine the zonal winds averaged over 10°S–10°N, and we estimate the driving of the SAO by gravity waves from the residual term ("missing drag") in the transformed Eulerian mean (TEM) zonal-average momentum budget (e.g., Andrews et al., 1987; Alexander and Rosenlof, 1996). We also investigate the SAO in quasi-geostrophic zonal winds derived from satellite observations of the MLS and the SABER satellite instruments, and in the winds directly observed by the TIMED Doppler Interferometer (TIDI) satellite instrument. Both SABER and TIDI are on the Thermosphere-Ionosphere-Mesosphere

Energetics and Dynamics (TIMED) satellite. Further, we investigate the gravity wave driving of the SAO based on absolute gravity wave momentum fluxes and absolute values of gravity wave drag derived from SABER satellite observations, and a correlation analysis between zonal winds and absolute gravity wave drag is carried out to reveal details of the SAO gravity wave driving.

The manuscript is organized as follows: Section 2 gives a description of the four reanalyses used in our study, and Sect. 3

gives a description of the instruments that provided the satellite data used in our study. In Sect. 4 we discuss the reanalysis SAO zonal winds and the SAO gravity wave driving expected from the reanalysis zonal momentum budget. Section 5 shows how the SAO is seen in the zonal winds determined from satellite data (quasi-geostrophic winds, as well as direct wind observations by TIDI), and Sect. 6 discusses the driving of the SAO based on SABER observations of absolute gravity wave momentum fluxes and absolute gravity wave drag. A correlation analysis is carried out in Sect. 7 to investigate the relation between SABER

absolute gravity wave drag and the SAO in more detail, and in Sect. 8 a similar correlation analysis is carried out for the reanalyses. Finally, Sect. 9 gives a summary of the paper.

## 2   Reanalysis data

In this paper four different meteorological reanalyses are used. The reanalysis ERA-Interim (see also Dee et al., 2011) of the European Centre for Medium-Range Weather Forecasts (ECMWF) has a horizontal model resolution of T255, corresponding

to a longitudinal grid spacing of ∼79 km at the equator. It uses 60 levels in the vertical with a model top level at 0.1 hPa, i.e. somewhat above the stratopause (see also Fig. 1). A parameterization of orographic gravity waves after Lott and Miller (1997) is included. A parameterization for nonorographic gravity waves, however, is missing and only included in later ECMWF model versions (see also Orr et al., 2010). To avoid reflection of model-resolved waves at the model top artificial damping





(Rayleigh friction) is used at pressures lower than $10\,\mathrm{hPa}$ (altitudes above $\sim 32\,\mathrm{km}$). For a summary of different reanalyses see
also, for example, Fujiwara et al. (2017) and Martineau et al. (2018).

The Japanese 55-year Reanalysis (JRA-55) (see also Kobayashi et al., 2015) of the Japanese Meteorological Agency (JMA) has a finer grid spacing with a horizontal resolution of T319 ($\sim 55\,\mathrm{km}$ at the equator). Like ERA-Interim, JRA-55 uses 60 model levels with the model top level at $0.1\,\mathrm{hPa}$ (cf. Fig. 1), a parameterization of orographic gravity waves is included (Iwasaki et al., 1989a,b), but no parameterization for nonorographic gravity waves. Rayleigh damping is applied at pressures
below $50\,\mathrm{hPa}$ (altitudes above $\sim 21\,\mathrm{km}$). In addition, the horizontal diffusion coefficient is gradually increased with altitude at pressures lower than $100\,\mathrm{hPa}$.

Unlike ERA-Interim and JRA-55, the Modern-Era Retrospective Analysis for Research and Applications, Version 2 (MERRA-2) reanalysis (see also Gelaro et al., 2017) uses 72 layers in the vertical with a model top at $0.01\,\mathrm{hPa}$, and a top layer mid level at $0.015\,\mathrm{hPa}$ ($\sim 78\,\mathrm{km}$) in the upper mesosphere. The horizontal resolution is $0.5°$ latitude $\times$ $0.625°$ longitude. Parameteriza-
tions for both orographic (McFarlane, 1987) and nonorographic gravity waves (Garcia and Boville, 1994; Molod et al., 2015) are included. Additional damping is applied at pressures less than $0.24\,\mathrm{hPa}$ (altitudes above $\sim 58\,\mathrm{km}$), i.e. at altitudes much higher than in ERA-Interim and JRA-55. One peculiarity of MERRA-2 is that, starting in August 2004, MLS temperature data are assimilated. This means that MERRA-2 is constrained by observations even in the mesosphere, while other reanalyses usually do not include observations above the stratopause. Further, the MERRA-2 nonorographic gravity wave drag scheme
was optimized for a better representation of the QBO and the SAO in the tropics (Molod et al., 2015).

Similar to MERRA-2, the ECMWF reanalysis ERA-5 (see also Hersbach and Dee, 2016; Hersbach et al., 2018, 2019, 2020) has a high model top with the top level at $0.01\,\mathrm{hPa}$ ($\sim 80\,\mathrm{km}$). The number of model levels is 137, resulting in a better vertical resolution than for all reanalyses previously described, including MERRA-2 (Fig. 1). The horizontal resolution is T639, according to a longitudinal grid spacing of $\sim 31\,\mathrm{km}$ at the equator. In our work we use the updated version ERA5.1
that uses an improved assimilation scheme for the period 2000–2006 (Simmons et al., 2020). ERA-5 uses parameterizations for orographic (Lott and Miller, 1997; Sandu et al., 2013) and nonorographic (Orr et al., 2010) gravity waves, but does not assimilate MLS data. The sponge layer starts at pressures lower than $10\,\mathrm{hPa}$ (altitudes above $\sim 32\,\mathrm{km}$) and depends on model level and zonal wavenumber in order to damp vertically propagating waves (e.g., Polichtchouk et al., 2017). An additional sponge layer starts at pressures lower than $1\,\mathrm{hPa}$ (altitudes above $\sim 48\,\mathrm{km}$). Unlike ERA-Interim, no Rayleigh friction is
applied at pressures lower than $10\,\mathrm{hPa}$. For comparison, Fig. 1 illustrates the model levels used in the different reanalyses for the altitude range of 30 to $90\,\mathrm{km}$ covered in this study.

## 3 The satellite instruments MLS, SABER, and TIDI

The Microwave Limb Sounder (MLS) is one of the instruments onboard the NASA satellite Aura. MLS is a limb sounding radiometer that observes atmospheric microwave emissions (e.g., Waters et al., 2006; Livesey et al., 2017). From these limb
observations, atmospheric temperature and a number of trace species are derived. In our study we use MLS version 4.2 atmospheric temperatures and geopotential height, which are available from the middle troposphere to the mesopause region


(pressures from 316 to 0.001 hPa). The vertical resolution is between ∼4 km in the stratosphere and ∼14 km around the mesopause. A detailed description of the temperature/pressure retrieval is given, for example, in Schwartz et al. (2008). The Aura satellite is in a sun-synchronous orbit. Therefore, MLS observations are always at two fixed local solar times. In the tropics, these local times are about 13:45 local solar time (LST) for the ascending orbit parts (i.e., when the satellite is flying northward) and 01:45 LST for the descending orbit parts (i.e., when the satellite is flying southward), according to the satellite equator crossing times. Measurements of MLS started on 8 August 2004 and are still ongoing at the time of writing.

The Sounding of the Atmosphere using Broadband Emission Radiometry (SABER) instrument was launched onboard the Thermosphere-Ionosphere-Mesosphere Energetics and Dynamics (TIMED) satellite in December 2001. SABER measurements started on 25 January 2002 and are still ongoing at the time of writing. TIMED has been approved to operate for three more years, until September 2023. Another three more years of operations will be proposed in near future. SABER is a broadband radiometer that observes atmospheric infrared emissions in limb-viewing geometry with an altitude resolution of about 2 km. Atmospheric temperatures are derived from infrared emissions of carbon dioxide ($CO_2$) at around 15 $\mu$m. The SABER temperature-pressure retrieval is described in detail by Remsberg et al. (2004) and Remsberg et al. (2008). More details on the SABER instrument are given, for example, in Mlynczak (1997) and Russell et al. (1999). In our study we use SABER version 2 temperatures, and in Sect. 6.1 we briefly introduce the method how absolute gravity wave momentum fluxes and absolute gravity wave drag can be derived from these temperature observations.

The TIMED satellite orbit is slowly precessing with a period of about 120 days. To make sure that always the same side of the satellite stays in the dark, TIMED performs yaw maneuvers approximately every 60 days. Accordingly, the local solar time of the satellite observations slowly drifts over one of the ∼60-day periods, and then jumps when a satellite yaw is performed. In Fig. 2 this is illustrated for the equatorial local solar times of SABER observations during ascending (black diamonds) and descending (black crosses) orbit parts, respectively, for the time period 2002 until 2018.

Since launch, the TIMED spacecraft has been decreasing in altitude by about 1 km per year. The inclination of the spacecraft has remained stable at 74°. However, the change in altitude has resulted in a drift of local time sampling, and hence, of the yaw date. The first TIMED yaw was in January 2002. At the time of writing, that yaw is now occurring in late December. As a consequence, the local time sampled in a given day or month changes every year. This effect could affect trend studies, but should not impact our work.

Another instrument onboard the TIMED satellite is the TIMED Doppler Interferometer (TIDI). Detailed information about TIDI can be found, for example, in Killeen et al. (2006) or Niciejewski et al. (2006). The TIDI instrument is a Fabry-Perot interferometer that was designed to observe atmospheric winds in the altitude range 70–120 km with an altitude resolution of about 2 km. This is achieved by using four individual telescopes to observe atmospheric emissions of rotational lines in the molecular oxygen ($O_2$) (0-0) band around 762 nm in limb-viewing geometry. One pair of telescopes is located on the sunlit side of the TIMED satellite (warm side), the other pair is located on the dark side (cold side). In each pair, one telescope views forward at an angle of 45° with respect to the satellite velocity vector, the other telescope views 45° rearward. In this way, the same air volume is observed by the two telescopes of a pair with a time difference of only 9 minutes. Based on these orthogonal measurements, wind vectors can be derived from the Doppler-shift of the atmospheric emissions. The wind vector





observations form two tracks on either side of the spacecraft, i.e. the warm side and the cold side, respectively. Figure 2 shows that these two tracks are at different local solar times with the local solar time of the cold side track differing from the local solar time of the corresponding SABER observations by only about half an hour. This is visible in Fig. 2 where local solar

times of TIDI equatorial observations are indicated by red diamonds (warm, ascending), red crosses (warm, descending), blue diamonds (cold, ascending), and blue crosses (cold, descending), respectively. Like for SABER, also TIDI observations are still ongoing at the time of writing.

## 4   The SAO in the ERA-Interim, JRA-55, ERA-5, and MERRA-2 reanalyses

### 4.1   Zonal wind

In our study, we focus on the 2002–2018 period during which gravity wave observations by the SABER instrument are available. From the reanalyses, we use global distributions of meteorological fields at 00:00, 06:00, 12:00, and 18:00 UT. For comparison with SABER data, we calculate values of the zonal wind averaged over 7 days and over the latitude band 10°S–10°N. Values are calculated in steps of 3 days, i.e. the time periods used for averaging are overlapping.

    Figure 3 shows zonal winds averaged over 10°S–10°N for the 2002–2018 time period for ERA-Interim, as well as the multi-

year average for the time period 2002–2018. For comparison, Fig. 3 shows in the lower right panel also the zonal wind of the SPARC zonal wind climatology, averaged over the latitude band 10°S–10°N. The same comparison is repeated in Figs. 4–6 for the JRA-55, ERA-5, and MERRA-2 reanalyses, respectively.

### 4.1.1   The stratopause SAO

All reanalyses capture some basic features of the SAO in the stratopause region and in the lower mesosphere. In all reanal-

yses, the first SAO period of a given year has the larger amplitude, as expected from observations (e.g., Garcia et al., 1997; Swinbank and Ortland, 2003). It is noteworthy that, while there is strong interannual variability in all reanalyses, this variability differs strongly among the different reanalyses. There are also other significant differences. For example, in ERA-Interim, the eastward winds of the first SAO period of a given year are somewhat stronger than in JRA-55, or in MERRA-2. Further, ERA-5 eastward jets are generally too strong at altitudes above ∼45 km, consistent with previous studies (Hersbach et al., 2018;

Shepherd et al., 2018). These overly strong eastward winds are caused by severe tapering of vorticity errors in the mesosphere, and this issue has been resolved from the introduction of IFS cycle 43r3 (11 July 2017) (Hersbach et al., 2018).

    Generally, large differences at high altitudes result because ERA-Interim and JRA-55 have lower model tops and introduce stronger artificial damping at lower altitudes than in MERRA-2 and ERA-5. Therefore, ERA-Interim winds strongly weaken at altitudes above 50 km, which, however, is less the case for JRA-55.

Next, we compare the four reanalyses with the SPARC zonal wind climatology (lower right panel in Figs. 3 to 6, respectively). The SPARC wind is a monthly climatology (Swinbank and Ortland, 2003; Randel et al., 2002, 2004) that combines wind observations by the Upper Atmosphere Research Satellite (UARS) instrument High Resolution Doppler Imager





(HRDI) (cf. Hays et al., 1993) and model data to interpolate gaps. Effects of the QBO and of tides have been widely removed (Randel et al., 2002). Compared to the SPARC climatology, the SAO in all four reanalyses has a larger amplitude in the up-
per stratosphere. Partly, this is caused by the fact that the SPARC climatology has only a monthly temporal resolution and will therefore smear out rapid temporal changes like the SAO. In addition, the SPARC climatology is based on an average of the years 1992–1998 (with reduced coverage after mid 1996), and the SAO in the upper stratosphere might have changed on average.

### 4.1.2 The SAO in the mesosphere, and the MSAO

At altitudes above ∼60 km, deviations between the SPARC climatology and the reanalyses become large. In the SPARC climatology at altitudes between 60 and 70 km, the zonal wind is continuously eastward, which, on average, is only the case in ERA-5. In ERA-5, however, eastward directed winds in this altitude range are often too strong.

Those eastward directed winds around 60 and 70 km altitude seem to be a real feature in climatological averages. For example, continuously eastward winds at the equator have been observed around 0.1 hPa (∼65 km) from October 2009 until
April 2010 by the Superconducting Submillimeter-Wave Limb-Emission Sounder (SMILES) instrument (Baron et al., 2013). During this period also in MERRA-2 eastward winds are seen around ∼65 km, but not in a multi-year average. Also multi-year averages of quasi-geostrophic winds that are derived from satellite observations and interpolated to the tropics show persistent eastward winds around ∼65 km. There is, however, strong interannual variability, and in several years it is observed that the zonal winds at altitudes around ∼65 km alternate between eastward and westward due to the SAO (see Smith et al. (2017) and
Sects. 5.2 and 5.3).

Another important feature in the SPARC climatology is a mesopause SAO that is in an anti-phase relation with the SAO at lower altitudes (see also, for example, Burrage et al., 1996) and has its peak amplitude around ∼80 km. Of course, the MSAO is not captured by ERA-Interim and JRA-55 because of their low model tops. Also MERRA-2 does not capture the MSAO; Due to a strong sponge layer the zonal wind in MERRA-2 is gradually damped to near zero close to the model top. Only ERA-5
partly captures the MSAO, and the wind reverses to westward at altitudes around 70 km, i.e. near the model top.

### 4.2 Estimates of gravity wave drag from reanalyses

Based on the transformed Eulerian mean (TEM) zonal mean momentum budget an expected value of the zonal-mean zonal gravity wave drag can be estimated from reanalyses. The zonal mean momentum equation is given by

$$\frac{\partial \overline{u}}{\partial t} + \overline{v}^* \left( \frac{(\overline{u}\cos\phi)_\phi}{a\cos\phi} - f \right) + \overline{w}^* \overline{u}_z = \overline{X}_{PW} + \overline{X}_{GW} \tag{1}$$

Here, $\overline{u}$ is the zonal-mean zonal wind, $\partial \overline{u}/\partial t$ the zonal wind tendency, $\overline{v}^*$ and $\overline{w}^*$ are the TEM meridional and vertical wind, respectively, $f$ is the Coriolis frequency, $a$ the Earth's radius, and $\phi$ the geographic latitude. $\overline{X}_{PW}$ and $\overline{X}_{GW}$ are the zonal-mean zonal wave drag due to global-scale waves and gravity waves, respectively. Subscripts $\phi$ and $z$ stand for differentiation in meridional and vertical direction, respectively. Overbars indicate zonal averages.





All terms in Eq. (1) except for $\overline{X}_{GW}$ can be calculated from the resolved meteorological fields of the reanalysis. The resolu-
tion (both horizontally and vertically) of the general circulation models used in the reanalyses, however, is too coarse to properly
resolve all scales of gravity waves. This means that part of the gravity wave spectrum is not resolved by the models, and ampli-
tudes of resolved gravity waves are usually underestimated (e.g., Schroeder et al., 2009; Preusse et al., 2014; Jewtoukoff et al.,
2015). Therefore, free-running general circulation models and reanalyses utilize parameterizations to simulate the contribu-
tion of gravity waves to the momentum budget (e.g., Fritts and Alexander, 2003; Kim et al., 2003; Alexander et al., 2010;
Geller et al., 2013).

Unlike those of free-running models, the meteorological fields of reanalyses are constrained by assimilation of numerous
observations. Where constrained by observations, the meteorological fields of reanalyses can be assumed to be quite realistic.
Under this assumption, the contribution $\overline{X}_{GW}$ in Eq. (1) can be calculated from the residual term ("missing drag") remaining
after quantifying all other contributions from the model-resolved fields (e.g., Alexander and Rosenlof, 1996; Ern et al., 2014,
265    2015).

Like in Ern et al. (2015), we calculate the zonal-mean zonal wave drag $\overline{X}_{res}$ due to waves that are resolved by the model
from the divergence of the Eliassen-Palm flux (EP-flux). Further, we assume that the zonal drag due to global-scale waves can
be approximated based on the resolved flux at zonal wavenumbers $k$ lower than 21:

$$\overline{X}_{PW} = \overline{X}_{res}(k < 21) \tag{2}$$

Under this assumption, our estimate of zonal mean gravity wave drag $\overline{X}_{GW}$ comprises the drag of model-resolved waves at
zonal wavenumbers higher than 20 ($\overline{X}_{res}(k > 20)$), gravity wave drag that is parameterized in the model ($\overline{X}_{param}$), and the
remaining imbalance ($\overline{X}_{imbalance}$) in the momentum budget that is caused by, for example, data assimilation:

$$\overline{X}_{GW} = \overline{X}_{res}(k > 20) + \overline{X}_{param} + \overline{X}_{imbalance} \tag{3}$$

with the "missing drag" consisting of the sum of $\overline{X}_{param}$ and $\overline{X}_{imbalance}$.

**4.2.1   ERA-Interim and JRA-55**

Figures 7 and 8 show the estimated gravity wave drag for ERA-Interim and JRA-55, respectively, averaged over the latitude
band 10°S–10°N. Similar as in Fig. 3, the gravity wave drag is given for the single years 2002 until 2018, as well as averaged
over these years (lower right panel in Figs. 7 and 8, respectively). Figures 7 and 8 show that gravity wave drag in the altitude
range 45–55 km is usually directed eastward, contributing to the driving of the eastward phase of the stratopause SAO with
a maximum value of about $5\,\mathrm{m\,s}^{-1}\mathrm{day}^{-1}$. Westward gravity wave driving in the stratopause region is much weaker and, on
average, does not contribute much to the driving of the stratopause SAO. This asymmetry has been pointed out before for
ERA-Interim by Ern et al. (2015). At high altitudes eastward gravity wave drag strongly increases, which is likely not realistic
and an effect of the sponge layer close to the model tops. This increase is most obvious above ∼55 km for ERA-Interim, and
above ∼45 km for JRA-55. Still, even though not very physical, the sponge layer effect seems to help simulate a more realistic
SAO (Polichtchouk et al., 2017). Switching off the sponge leads to stronger mesospheric eastward winds at the equator.





### 4.2.2 MERRA-2

Analogously to ERA-Interim and JRA-55, Fig. 9 shows that the MERRA-2 gravity wave driving in the altitude region 45–55 km (around the stratopause) is prevalently directed eastward with peak values of about $7\,\mathrm{m\,s^{-1}day^{-1}}$, i.e. stronger than in ERA-Interim and JRA-55. Westward directed gravity wave drag in the stratopause region is generally weaker with peak values

of usually $\sim 2\,\mathrm{m\,s^{-1}day^{-1}}$.

In the stratosphere, the QBO westward and eastward phases are usually stacked, and, since the zonal wind is usually stronger during QBO westward phases than during QBO eastward phases, the range of westward gravity wave phase speeds encountering critical level filtering is usually larger than the range of eastward phase speeds. This will lead to an asymmetry of the gravity wave spectrum with a larger amount of eastward momentum flux entering the stratopause region and the mesosphere,

and, consequently, to the prevalently eastward driving of the stratopause SAO by gravity waves.

At times, the QBO eastward and westward phases are not perfectly stacked, resulting in less pronounced asymmetric wave filtering by the QBO. This is the case, for example, during April to June 2006 and April to June 2013. During these periods we find also relatively strong westward directed gravity wave drag in the stratopause region (around 50 km altitude), and these enhancements seem to contribute to the formation of stronger downward propagating SAO westward phases. Indications for

the less asymmetric filtering of the gravity wave spectrum during 2006 were also found before from satellite observations (Ern et al., 2015).

Different from ERA-Interim and JRA-55, MERRA-2 assimilates MLS observations in the mesosphere. Further, the MERRA-2 model top is at higher altitudes, and increased damping is used only above $\sim 58$ km. Therefore, reasonable estimates of gravity wave drag should also be possible in the middle mesosphere. It is striking that in the altitude range 55 to somewhat above 65 km

westward gravity wave drag is increased compared to the stratopause region, and sometimes is as strong as eastward gravity wave drag. In this altitude range, the westward gravity wave drag often contributes to the closure of the mesospheric SAO eastward wind jet at its top. Nevertheless, in this altitude range, the westward gravity wave drag is still, on average, only about half as strong as eastward gravity wave drag as shown from the multi-year average (Fig. 9, lower right panel). At altitudes above $\sim 65$ km there is a sudden increase of eastward gravity wave drag in MERRA-2, which is likely unrealistic and related

to damping in the sponge layer close to the model top, similar as in ERA-Interim and JRA-55.

Note that MERRA-2 gravity wave drag is more strongly linked to vertical gradients of the background wind than is the case for ERA-Interim and JRA-55. Different from ERA-Interim and JRA-55, MERRA-2 uses a nonorographic gravity wave drag scheme. This scheme was additionally tuned to improve the QBO and the SAO in the tropics (Molod et al., 2015). Therefore, the strong link between gravity wave drag and vertical gradients of the background wind could be an effect of the dedicated

tuning of this gravity wave drag parameterization. This effect will be investigated in more detail in Sect. 7 based on satellite data, and in Sect. 8 for the reanalyses.





### 4.2.3 ERA-5

Like ERA-Interim, JRA-55, and MERRA-2, the ERA-5 reanalysis shows an asymmetry between eastward and westward gravity wave drag in the stratopause region (Fig. 10). However, peak values of eastward gravity wave drag are somewhat lower than those of MERRA-2. Furthermore, in the stratopause region, enhanced values of gravity wave drag are not as closely linked to zonal wind vertical gradients as it is the case for MERRA-2. This finding is surprising because, like MERRA-2, ERA-5 contains a nonorographic gravity wave drag scheme. Possibly, this difference is caused by different settings of the gravity wave drag schemes. For instance, enhanced gravity wave momentum fluxes were introduced in the tropics to improve the representation of the QBO and the SAO in MERRA-2 (Molod et al., 2015), which is different in ERA-5.

The ERA-5 characteristics change at altitudes above about 65 km. At these altitudes also in ERA-5 enhanced gravity wave drag is closely linked to zonal wind vertical gradients, and strong westward directed gravity wave drag contributes to the reversal of the mesospheric eastward directed winds and the formation of the mesopause SAO, qualitatively consistent with MERRA-2. In MERRA-2, however, there is no clear wind reversal. Possibly, the sponge layer in MERRA-2 is stronger than that in ERA-5, preventing the formation of a clear MSAO. Still, there is some eastward directed gravity wave drag near the model top in ERA-5 that seems to be related to the model sponge layer, but that is much weaker than in MERRA-2.

## 5 The SAO as seen in satellite data

### 5.1 Interpolated quasi-geostrophic winds in the tropics

Following the approach used in previous studies (e.g., Oberheide et al., 2002; Ern et al., 2013; Smith et al., 2017; Sato et al., 2018), quasi-geostrophic winds can be calculated from the geopotential fields derived from satellite soundings. For stationary conditions, and neglecting the drag exerted by atmospheric waves, the zonal and meridional momentum equations can be written as follows

$$-\left(f + \frac{u \tan \phi}{a}\right) v + \frac{1}{a \cos \phi} \frac{\partial \Phi}{\partial \lambda} = 0 \tag{4}$$

$$\left(f + \frac{u \tan \phi}{a}\right) u + \frac{1}{a} \frac{\partial \Phi}{\partial \phi} = 0 \tag{5}$$

Here, $u$ and $v$ are the zonal and the meridional wind, respectively, $a$ the Earth radius, $\phi$ the geographic latitude, and $\Phi$ the geopotential. For further details see Andrews et al. (1987), Oberheide et al. (2002), or Ern et al. (2013). These equations can be easily solved for $u$ and $v$.

The quasi-geostrophic approach gives good results in the extratropics, but is not reliable in the tropics because the Coriolis parameter is close to zero. Recently, it has been shown by Smith et al. (2017) that an interpolation of the quasi-geostrophic zonal wind starting from $10°$S and $10°$N can be used as a proxy for the zonal wind at the equator and is in good agreement with wind observations by lidar below about 80 km.



As direct wind observations in the tropical mesosphere are sparse, we will also make use of this approach, even though interpolated quasi-geostrophic winds will still be affected by biases. In order to make sure that our findings are robust, we will use a number of different zonal wind data sets in Sects. 6 and 7 to check whether our findings of the SAO gravity wave driving hold for different choices of background winds.

For our study, we utilize zonal-average quasi-geostrophic zonal winds calculated for time intervals of three days with a time step of three days, i.e. the time windows used for calculating the winds are non-overlapping. This data set has been previously used for studies in the extratropics (Ern et al., 2013, 2016; Matthias and Ern, 2018). For studying the interaction of gravity waves with the SAO zonal wind in the latitude band 10°S–10°N, we use the average of the quasi-geostrophic wind at 12°S and 12°N as a proxy for the zonal wind in this latitude band at altitudes above 45 km, similar as in Smith et al. (2017). At lower altitudes, reanalysis winds should be more reliable, so we do not use quasi-geostrophic winds at altitudes below 35 km. Instead, we use the ERA-Interim winds already presented in Fig. 3, and a smooth transition between ERA-Interim and quasi-geostrophic winds derived from SABER or MLS satellite observations in the altitude range 35–45 km.

### 5.2 MLS quasi-geostrophic winds

For particular comparison with the MERRA-2 reanalysis that assimilates MLS data, Fig. 11 shows zonal winds for the merged data set of ERA-Interim and interpolated MLS quasi-geostrophic winds. To reduce the effect of tides, MLS winds are calculated from an average over ascending and descending orbit branches, i.e. data from the two MLS equator crossing times are averaged. As MLS observations started in mid-2004, the panels for the years 2002 and 2003 are left blank.

Figures 6 and 11 show that at altitudes below ∼60 km MLS and MERRA-2 winds are very similar. On the one hand, this is expected because MLS data are assimilated in MERRA-2. On the other hand, this shows that our interpolated quasi-geostrophic winds are useful in the tropics. Still, these interpolated winds are not considered to be reliable at altitudes above ∼75 km. This is indicated, for example, by the more eastward winds and the overly short duration of the SAO westward wind phases at altitudes above ∼70 km when compared with the SPARC climatology. This bias is possibly an effect of tides. Although both ascending and descending nodes enter the estimation of MLS quasi-geostrophic winds, it is not expected that tidal effects will completely cancel out.

### 5.3 Merged SABER quasi-geostrophic and TIDI wind observations

So far we have discussed wind data sets of four reanalyses, as well as interpolated quasi-geostrophic winds based on MLS observations. Another main purpose of our work is to study the interaction of SABER gravity wave observations with the background wind. Of course, both the SAO and tides contribute to the variations of the winds in the tropics. As shown in Fig. 2, the local solar times of SABER equator crossings slowly change over time. Therefore, it is important to compare gravity wave observations and winds observed at the same local solar times.

For this purpose, we have composed a combined data set of SABER quasi-geostrophic winds in the altitude range 45–75 km, ERA-Interim winds below 35 km, and a smooth transition between ERA-Interim and SABER winds in the altitude range 35–45 km. At altitudes above ∼80 km we use directly observed TIDI "cold side" winds. As shown in Fig. 2, the local solar time





of TIDI cold side winds matches the local solar times of SABER observations better than about half an hour. Winds in the gap between 75 and 80 km are interpolated. Similar as in the study of Dhadly et al. (2018), we omit less reliable TIDI data from periods when the angle $\beta$ between orbital plane and the Earth-Sun vector exceeds 55°, i.e. when the TIMED orbital plane is near the terminator. Data gaps that are caused by omitting these data, as well as other data gaps that are shorter than 40 days are closed by linear interpolation in time. A larger data gap from November 2016 until March 2017 is closed by using interpolated

SABER quasi-geostrophic winds also at altitudes above 75 km. Interpolated SABER quasi-geostrophic winds are used above 75 km also before April 2002, because TIDI cold side winds are available only after that date.

Figure 12 shows these combined winds. SABER and TIDI winds were averaged over ascending and descending TIMED satellite equator passings, i.e., they represent an average over different local solar times. At altitudes below ∼70 km these winds are very similar to those derived from MLS (see Fig. 11). Although ascending and descending orbit data are combined,

there are notable variations that are related to the 60-day yaw cycle of the TIMED satellite and the corresponding changes in the local solar time of SABER and TIDI observations. This shows the importance of selecting wind data at the correct local solar time, particularly at higher altitudes.

The main difference between Fig. 11 and Fig. 12, however, are the winds at altitudes above 80 km where TIDI wind observations are used. On average, the TIDI winds are more westward than the quasi-geostrophic winds derived from MLS, and even

somewhat more westward than the SPARC climatology (Fig. 12, lower right). Particularly the maxima of both SAO eastward phases at altitudes above around 85 km are less pronounced. Because at altitudes above 80 km variations that are linked to the TIMED yaw cycles and the corresponding changes in local solar time are quite strong, this could be an effect of tides. The TIDI instrument samples atmospheric tides at the same phase as SABER. Since wind variations due to tides can be of the same magnitude as variations due to the SAO, the combined data set of SABER and TIDI winds should therefore be the best choice

for representing the atmospheric background conditions relevant for SABER gravity wave observations.

A more comprehensive analysis of tides based on TIDI winds has been carried out in previous studies (e.g., Oberheide et al., 2006; Wu et al., 2011; Dhadly et al., 2018). An in-depth investigation of the effect of tides on the distribution of gravity waves, however, is beyond the scope of our study. Overall, the differences between the different wind data sets show the importance of further global wind observations in the upper mesosphere and lower thermosphere, and particularly in the tropics. As there are

notable differences between different wind data sets, in Sect. 7 we will compare SABER gravity wave observations to several different wind data sets in order to find out which findings are robust and widely independent of the wind data used.

## 6  Satellite observations of the SAO driving by gravity waves

One of the key parameters that is relevant for the interaction of gravity waves with the background flow is the vertical flux of gravity wave pseudomomentum ($\mathbf{F}_{ph}$), denoted in the following as "gravity wave momentum flux". The momentum flux of a

gravity wave is given as:

$$\mathbf{F}_{ph} = (F_{px}, F_{py}) = \varrho \left( 1 - \frac{f^2}{\widehat{\omega}^2} \right) (\overline{u'w'}, \overline{v'w'}) \tag{6}$$





with $F_{px}$ and $F_{py}$ the gravity wave momentum flux in zonal and meridional direction, respectively, $\varrho$ the atmospheric density, $f$ the Coriolis frequency, $\widehat{\omega}$ the intrinsic frequency of the gravity wave, and $(u', v', w')$ the vector of zonal, meridional and vertical wind perturbations due to the gravity wave (e.g., Fritts and Alexander, 2003). If a gravity wave propagates conservatively, the momentum flux of a gravity wave stays constant. However, if a gravity wave dissipates while propagating upward, momentum flux is no longer conserved, and the gravity wave exerts drag on the background flow. This drag $(X, Y)$ is related to the vertical gradient of momentum flux:

$$(X, Y) = -\frac{1}{\varrho} \frac{\partial \mathbf{F}_{ph}}{\partial z} \tag{7}$$

with $X$ and $Y$ the gravity wave force in zonal and meridional direction, respectively, and $z$ the vertical direction. As will be explained in the next subsection, gravity wave momentum flux can also be derived from temperature observations of satellite instruments.

### 6.1 Estimates of absolute gravity wave momentum fluxes and drag from SABER observations

For deriving gravity wave momentum fluxes from temperature altitude profiles observed by SABER, we make use of the method described in our previous studies (Ern et al., 2004, 2011; Ern et al., 2018). First, the atmospheric background temperature is estimated, separately for each altitude profile. This estimate consists of the zonal average temperature profile. Further, 2D zonal-wavenumber / wave-frequency spectra are determined from SABER temperatures for a set of latitudes and altitudes. Based on these spectra, the contribution of global-scale waves is calculated at the location and time of each SABER observation. Both zonal average profile and global scale waves are removed from each altitude profile.

For our study, it is important that this 2D spectral approach is capable of effectively removing all global-scale waves that are important in the tropics, such as inertial instabilities in the tropical stratosphere and stratopause region (e.g., Rapp et al., 2018; Strube et al., 2020), and different equatorial wave modes in the stratosphere (e.g., Ern et al., 2008) and in the mesosphere and mesopause region (e.g., Garcia et al., 2005; Ern et al., 2009). In particular, Kelvin waves contribute significantly to the temperature variances in the tropics and are difficult to remove by other techniques because they can have very short wave periods, and their vertical wavelengths are in the same range as that of small scale gravity waves. Each altitude profile is additionally high-pass filtered to remove fluctuations of vertical wavelengths longer than about $25\,\mathrm{km}$ to focus on those gravity waves that are covered by our momentum flux analysis, and to remove remnants of global-scale waves. Further, we explicitly remove tides by removing offsets and quasi-stationary zonal wavenumbers of up to 4, separately for ascending and descending orbit parts of SABER. In this way, we cover major tidal modes, such as the diurnal westward zonal wavenumber 1 (DW1), the semidiurnal westward zonal wavenumber 2 (SW2), and the diurnal eastward zonal wavenumber 3 (DE3). The final result of this procedure are altitude profiles of temperature fluctuations that can be attributed to small scale gravity waves.

As introduced by Preusse et al. (2002), for each altitude profile the amplitude, vertical wavelength $\lambda_z$, and the phase of the strongest wave component are determined in sliding 10–km vertical windows. Provided a close enough spacing in space and time, the gravity wave horizontal wavelength parallel to the satellite measurement track ($\lambda_{h,AT}$) can be estimated from pairs of consecutive altitude profiles if the same wave is observed with both profiles of a pair. To make sure that the same wave is





observed in both profiles of a pair, a vertical wavelength threshold is introduced and we assume that the same wave is observed if $\lambda_z$ differs between the two profiles by not more than 40%. Pairs with non-matching vertical wavelengths are discarded. This omission of pairs does not introduce significant biases in distributions of gravity wave squared amplitudes (e.g., Ern et al., 2018). Therefore the selected pairs should be representative of the whole distribution of gravity waves.

Taking $\lambda_{h,AT}$ as a proxy for the true horizontal wavelength $\lambda_h$ of a gravity wave, absolute values of gravity wave momentum flux $F_{ph}$ can be estimated:

$$F_{ph} = \frac{1}{2}\varrho \left(\frac{g}{N}\right)^2 \frac{\lambda_z}{\lambda_h} \left(\frac{\widehat{T}}{T}\right)^2 \tag{8}$$

with $g$ the gravity acceleration, $N$ the buoyancy frequency, $T$ the background temperature, and $\widehat{T}$ the gravity wave temperature amplitude (see also Ern et al., 2004).

Main error sources of $F_{ph}$ are the undersampling of short horizontal wavelength waves (aliasing), the overestimation of $\lambda_h$ by $\lambda_{h,AT}$ (see also, for example Preusse et al. (2009), Alexander (2015), Ern et al. (2017, 2018), or Song et al. (2018)), and the attenuation of gravity wave amplitudes by the instrument sensitivity function (see also, for example Preusse et al., 2002). The approximate SABER sensitivity function is given in Ern et al. (2018), and a comprehensive discussion of the observational filter of infrared limb sounders is given in Trinh et al. (2015). As was estimated by Ern et al. (2004) overall errors of $F_{ph}$ are large, at least a factor of two.

In situations of strong vertical shear of the background winds, a proxy of the absolute gravity wave forcing $XY$ on the background flow can be estimated from the vertical gradient of absolute gravity wave momentum flux:

$$XY = -\frac{1}{\varrho}\frac{\partial F_{ph}}{\partial z}. \tag{9}$$

For additional details please see Warner et al. (2005) and Ern et al. (2011). Similar to absolute gravity wave momentum fluxes, no directional information is available for $XY$, but it can often be assumed that gravity wave drag and the vertical gradient of the background wind have the same direction. Based on this assumption, meaningful results were obtained already in several studies in the extratropics (Ern et al., 2013, 2016) and, particularly, in the tropics for the QBO in the stratosphere (Ern et al., 2014) and the stratopause SAO (Ern et al., 2015). Similarly to Ern et al. (2015), our data sets of absolute gravity wave momentum fluxes and gravity wave drag are averages over 7 days with a step of 3 days, i.e., the time windows used for averaging are overlapping.

## 6.2 Effect of the background winds on SABER gravity wave momentum fluxes

First, we investigate how SABER absolute gravity wave momentum fluxes are modulated by the background winds. Figure 13 shows absolute gravity wave momentum fluxes observed by SABER, averaged over the latitude band 10°S–10°N. We also average over data from ascending and descending parts of the satellite orbit to reduce the effect of tides. Shown are distributions for the years 2002–2018, as well as an average over the whole 2002–2018 period (lower right in Fig. 13). Contour lines represent the combined data set of zonal winds from ERA-Interim, SABER quasi-geostrophic winds, and TIDI direct wind observations, as presented in Fig. 12.





Figure 13 shows that absolute gravity wave momentum flux in the stratopause region and in the middle mesosphere is usually strongest during periods of westward winds. This finding is consistent with the results obtained for the SSAO by Ern et al. (2015) and indicates that, due to the selective filtering of the gravity wave spectrum by the QBO in the stratosphere,

the gravity wave spectrum in the stratopause region and in the middle mesosphere is dominated by gravity waves of eastward directed phase speeds. An overall decrease of momentum fluxes with altitude shows that gravity waves dissipate gradually with increasing altitude. In addition to this overall decrease, momentum fluxes decrease more strongly in zones of eastward (positive) wind shear, which indicates that gravity waves interact with the SAO winds in the stratopause region and middle mesosphere and contribute to the driving of the SAO. This effect will be investigated in more detail in Sect. 6.3 based on

vertical gradients of absolute gravity wave momentum fluxes. In the upper mesosphere and in the mesopause region, there is no such clear relationship between momentum fluxes and positive wind shear. This effect will also be discussed later in Sect. 6.3.

### 6.3   Interaction of the SABER absolute gravity wave drag and the tropical zonal wind

Figure 14 shows SABER absolute gravity wave drag that is calculated from vertical gradients of SABER absolute gravity wave

momentum fluxes. Again, values are averaged over the latitude band $10°$S–$10°$N and over ascending and descending orbit data. Similarly to Fig. 13, distributions of absolute gravity wave drag are shown for each given year of the 2002–2018 period. The lower right panel in Fig. 14 is an average over the whole 2002–2018 period. Contour lines represent the zonal winds shown in Fig. 12. From Fig. 14 we can see that SABER absolute gravity wave drag generally increases with height from close to zero at 30 km to around $20\,\mathrm{m\,s^{-1}day^{-1}}$ between 80 and 90 km. It has a local maximum around 50 km with peak values of about

$1$–$2\,\mathrm{m\,s^{-1}day^{-1}}$, and another local maximum between around 80 and 85 km with peak values of about $30\,\mathrm{m\,s^{-1}day^{-1}}$. The first maximum is likely related to the SSAO, while the second maximum is likely related to the MSAO.

### 6.3.1   The SSAO and the SAO in the middle mesosphere

In the stratopause region, peak values of SABER absolute gravity wave drag are seen mainly during eastward wind shear, while values are much reduced during westward wind shear, indicating that gravity wave drag is mainly directed eastward and

contributes to the driving of the SAO eastward wind phases. This finding is consistent with the HIRDLS observations discussed by Ern et al. (2015) and becomes even clearer when looking at Figs. 15 and 16.

Figures 15 and 16 show for each year, as well as for the 2002–2018 average year, the SABER absolute gravity wave drag anomaly, i.e. for each year the absolute gravity wave drag is divided by the altitude-dependent annual mean. Overplotted contour lines in Fig. 15 represent the zonal winds shown in Fig. 12, while the contour lines in Fig. 16 represent the vertical

gradient of this zonal wind. Figures 15 and 16 reveal that SABER absolute gravity wave drag is enhanced mainly during eastward wind shear, not only in the stratopause region, but in the whole altitude range of about 40–70 km.

Parts of the gravity wave spectrum, particularly those of slow ground based phase speeds, have encountered critical levels already at lower altitudes by the QBO (cf. Ern et al., 2014, 2015) and cannot contribute to the SAO driving. Therefore, an enhancement of gravity wave drag mainly during eastward zonal wind shear does not necessarily mean that critical level





filtering of gravity waves is the only dominant process. Another effect of vertical wind shear, in addition to the formation of critical levels, is a reduction of intrinsic phase speeds for parts of the gravity wave spectrum and, thus, a reduction of gravity wave saturation amplitudes for this part of the spectrum. This means that wave saturation apart from critical levels, i.e. saturation of high ground based phase speed gravity waves, can also play an important role in the stratopause region, and even more at higher altitudes. Indications for the importance of saturation of high phase speed gravity waves for the SSAO were

indeed found by Ern et al. (2015) by investigating gravity wave momentum flux spectra observed from satellite.

In the stratopause region the magnitudes of SABER gravity wave drag (peak values of around $1$–$2\,\mathrm{m\,s^{-1}day^{-1}}$) are similar or even stronger than those obtained by model simulations of the SSAO (e.g., Richter and Garcia, 2006; Osprey et al., 2010; Peña–Ortiz et al., 2010; Smith et al., 2020) and similar to values derived from Rayleigh lidar observations (Deepa et al., 2006; Antonita et al., 2007). Comparison with the reanalyses gives a somewhat different picture: SABER gravity wave drag is usually

weaker than peak values of eastward gravity wave drag of the four reanalyses considered in our study. For example, at around $50\,\mathrm{km}$ altitude peak values of eastward gravity wave drag in the multi-year averages are around 3 to $4\,\mathrm{m\,s^{-1}day^{-1}}$ for ERA-Interim (cf. Fig. 7), 3 to $6\,\mathrm{m\,s^{-1}day^{-1}}$ for JRA-55 (cf. Fig. 7, but values could be already affected by the model sponge layer), $\sim 3\,\mathrm{m\,s^{-1}day^{-1}}$ for MERRA-2 (cf. Fig. 9), and $\sim 2\,\mathrm{m\,s^{-1}day^{-1}}$ for ERA-5 (cf. Fig. 10).

Generally, observations cover only parts of the whole spectrum of gravity waves and should therefore underestimate gravity

wave drag. An underestimation of the gravity wave drag derived from SABER observations would be expected for two reasons. First, SABER momentum fluxes are likely underestimated due to overestimation of derived horizontal wavelengths by undersampling of observed gravity waves (aliasing) and by adopting along-track wavelengths instead of the true horizontal wavelengths (cf. Ern et al., 2018, and references therein). Second, the SABER instrument is sensitive only to gravity waves of horizontal wavelengths longer than $100$–$200\,\mathrm{km}$ and does therefore not cover the whole spectrum of gravity waves. In par-

ticular, it is indicated that short horizontal wavelength convectively generated gravity waves that cannot be seen by SABER contribute significantly to the driving of the SSAO (e.g., Beres et al., 2005; Kang et al., 2018). For further discussion regarding the observational filter of the instrument please see Trinh et al. (2015).

In their study, Smith et al. (2020) conclude that free-running models would have difficulties to simulate a realistic SSAO because of insufficient gravity wave forcing. Taking into account the observational filter effect, this conclusion is supported by

the fact that SABER absolute gravity wave drag is similar to the gravity wave drag of free-running global models, but lower than that of the reanalyses.

### 6.3.2 Upper mesosphere: the MSAO

In the upper mesosphere, at altitudes between about $\sim 75\,\mathrm{km}$ and $80\,\mathrm{km}$, the clear relationship between eastward wind shear and absolute gravity wave drag apparently does not hold any longer (see Figs. 15 and 16). This is expected because the asymmetric

wind filtering effect of the gravity wave spectrum induced by the QBO in the stratosphere should gradually fade out. Instead, the wind filtering in the stratopause region and the middle mesosphere should become more relevant.

This is supported by the fact that the MSAO is approximately in anti-phase with the SAO at the stratopause and in the middle mesosphere. It is believed that this is anti-phase relationship is caused by the dissipation of gravity waves that are selectively





filtered by the winds in the middle mesosphere. Gravity waves that have phase speeds opposite to the prevailing wind direction
in the stratopause region and the middle mesosphere, consequently, have high intrinsic phase speeds and, thus, high saturation
amplitudes (see also Fritts, 1984; Ern et al., 2015, and references therein). When reaching the upper mesosphere, these waves
saturate and contribute to the wind reversal, resulting in the observed anti-correlation of SAO winds in the middle and the upper
mesosphere. This means that winds in the upper mesosphere are westward when they are eastward in the middle mesosphere,
and vice versa. Accordingly, in the upper mesosphere gravity waves are expected to contribute both to the MSAO eastward and
the MSAO westward winds.

Interestingly, as shown by the SPARC zonal wind climatology, the downward propagation with time of the MSAO eastward
and westward wind phases is much slower than the downward propagation of the eastward wind phase of the SSAO and the
SAO in the middle mesosphere. Therefore, the characteristics of the gravity wave forcing should also be different in these two
altitude ranges. This will be investigated in more detail in Sect. 7.

Peak values of SABER absolute gravity wave drag in the altitude range of 75–80 km are about $20$–$30 \, \mathrm{m \, s^{-1} day^{-1}}$ (see
Fig. 14). As stated before, due to the SABER observational filter, these values are expected to be likely a lower estimate of
the total gravity wave drag. Indeed, in the mesopause region, Lieberman et al. (2010) obtained gravity wave peak values of
typically around $100 \, \mathrm{m \, s^{-1}}$, estimated as residual drag from the momentum budget using TIMED observations of SABER and
TIDI. However, similar as for the simulation of the stratopause SAO, gravity wave drag peak values of model simulations are
much weaker. For example, Richter and Garcia (2006) or Peña–Ortiz et al. (2010) obtained peak values of gravity wave drag
of only around $10$–$20 \, \mathrm{m \, s^{-1} day^{-1}}$ in their simulations of the SAO in the altitude range 75–85 km.

### 6.3.3   The region above the MSAO

Also at altitudes above 80 km, there is no clear relationship between eastward wind shear and absolute gravity wave drag.
Moreover, compared to the altitude range 30–80 km, there is a structural change in the distribution of absolute gravity wave
drag (see Figs. 15 and 16).

In the whole lower altitude regime 30–80 km we find downward propagation of absolute gravity wave drag enhancements
with time. At altitudes ∼30–75 km this downward propagation is relatively steep and related to the zones of eastward directed
SAO wind shear. At altitudes 75–80 km we still find downward propagation, although much slower, and seemingly related to
the downward propagation rate of the SAO wind phases (cf. Fig. 12).

Conversely, in the upper altitude regime above 80 km, enhancements of absolute gravity wave drag propagate *upward* with
time. These variations are obviously not directly related to the SAO winds, but to the variations that are caused by the varying
local solar time of SABER observations. The variations of absolute gravity wave drag at altitudes above ∼80 km are caused by
tides that are sampled at different local solar times while the TIMED satellite orbit precesses. For upward propagating tides the
phase propagation is downward with time (e.g., Smith, 2012; Sridharan, 2019). However, due to orbit precession of the TIMED
satellite, the SABER sampling gradually shifts to earlier local solar times, as shown in Fig. 2. This leads to an apparent upward
phase propagation with time of observed tides and, accordingly, to the observed apparent upward propagation of gravity wave
drag maxima because gravity wave drag should be directly linked with the wind shear induced by the tides.





At high altitudes, an increasing influence of tides on the distribution of gravity waves would also be expected from previous findings. At some point, the effect of tides should dominate over the effect of the SAO. Figure 30 in Baldwin et al. (2001) shows

that the MSAO in the upper mesosphere has a sharp amplitude peak of $30\,\mathrm{m\,s^{-1}}$ at $80\,\mathrm{km}$ altitude. The MSAO amplitude drops below $\sim 10\,\mathrm{m\,s^{-1}}$ already below $90\,\mathrm{km}$. Simultaneously, the amplitude of tides increases with altitude. For example, Fritts et al. (1997) found amplitudes of $5$–$10\,\mathrm{m\,s^{-1}}$ below $75\,\mathrm{km}$, increasing to about $20\,\mathrm{m\,s^{-1}}$ at $90\,\mathrm{km}$ for diurnal tides observed by radar near the equator during August 1994. These values are roughly in agreement with simulations of the Canadian Middle Atmosphere Model (CMAM) (McLandress et al., 2002). Based on these findings, it would be expected that in the altitude range

$80$–$90\,\mathrm{km}$ there should be a transition between a regime that is mainly dominated by the MSAO around $80\,\mathrm{km}$, and another regime that is increasingly dominated by tides at higher altitudes. Worthy of remark, this is also reflected in the observed SABER absolute gravity wave drag distribution.

## 7   Correlation between the observed absolute gravity wave drag and tropical zonal wind

Next, we carry out a correlation analysis in order to quantify the robustness of the interaction between SABER absolute gravity

drag and tropical zonal background wind. To do so, we calculate the temporal correlation between absolute gravity wave drag and the vertical gradient of the background zonal wind as well as the temporal correlation between absolute gravity wave drag and absolute values of the zonal background wind. These correlations are calculated for fixed altitudes, separately for each given year of the 2002–2018 period, and for the distributions that are obtained by averaging over these years.

Figure 17 shows 2002–2018 averages of zonal winds (first column), SABER absolute gravity wave drag with zonal wind

contour lines (second column), normalized SABER gravity wave drag with contour lines of zonal wind vertical gradients (third column), correlation coefficients between SABER gravity wave drag and $du/dz$ for each year, as well as for the 2002–2018 average distribution (fourth column), and correlation coefficients between SABER gravity wave drag and $|u|$ for each year, as well as for the 2002–2018 average distribution (fifth column). In order to find out whether the results are robust and insensitive to details of a wind data set considered, the correlation analysis is carried out for different wind data sets.

### 7.1   Correlation between SABER absolute gravity wave drag and $du/dz$

From theoretical considerations (e.g., Hamilton and Mahlmann, 1988; Dunkerton and Delisi, 1997), from first satellite observations of the gravity wave driving of the SAO in the stratopause region (Ern et al., 2015), and from the findings in Sect. 6.3, we expect that in a certain altitude range gravity waves mainly contribute to the driving of the SAO eastward phases and their downward propagation with time. For this altitude range, it is expected that gravity wave drag should mainly act during

eastward wind shear.

To find out in which altitude range this is the case, we calculated for each year separately the temporal correlation between SABER absolute gravity wave drag and $du/dz$. Since only absolute values of gravity wave drag are available, some care has to be taken when interpreting these results. If a correlation coefficient is positive, this means that in a given year, at the altitude considered, the gravity wave forcing takes mainly place during eastward wind shear, and the forcing is very likely eastward.





Similarly, if the correlation coefficient is negative, the gravity wave forcing takes mainly place during westward wind shear, and the forcing is very likely westward. If the correlation is close to zero, this means that either the relationship between gravity wave drag and $du/dz$ is random, or eastward and westward forcing could be similarly strong.

### 7.1.1  Stratosphere (QBO)

Figure. 17, first row, shows the results for the combined data set of ERA-Interim, SABER, and TIDI winds, averaged over
ascending and descending orbit data. In the stratosphere between about 30 and 40 km, there is an alternating pattern of strong positive and weak correlations, which is likely not an effect of the SAO. In this altitude range the QBO is the dominant mode of tropical wind variability. It has been shown by Ern et al. (2014) that absolute HIRDLS and SABER gravity wave drag is stronger during eastward directed QBO wind shear. Although the 10 km vertical window used in our study is relatively coarse and will often average over the stacked zones of eastward and westward directed QBO wind shear, this asymmetry can lead to
the observed alternating pattern of correlations. Since the QBO averages out in the 2002–2018 multi-year average, correlations for the average distributions are generally weak at altitudes below about 45 km. Similar alternating patterns are also found for all other wind data sets presented in Fig. 17 that contain the QBO. Even for the correlation between the SPARC zonal wind climatology and absolute gravity wave drag an alternating pattern can be found in the stratosphere. However, this pattern is different because the QBO signal is only contained in the SABER absolute gravity wave drag, but not in the winds, as the same
wind climatology is assumed for all years.

### 7.1.2  Stratopause region (SSAO) and middle mesosphere

In all panels of the fourth column in Fig. 17, starting from about 45 km upward, correlation coefficients are mostly strongly positive for the quasi-geostrophic data sets based on the SABER observations (Fig. 17, first to sixth row), independently of the treatment of ascending and descending orbit data. This is also the case if MLS quasi-geostrophic winds (Fig. 17, seventh row),
or the SPARC wind climatology (Fig. 17, bottom row) are used as background winds.

For the SABER and MLS quasi-geostrophic wind data sets (Fig. 17, rows 1–7) the altitude range of positive correlations is from about 45 km to 75–80 km, indicating that in this altitude range gravity waves mainly contribute to the driving of the eastward SAO phase. Interestingly, for the SPARC wind climatology (Fig. 17, bottom row) the altitude range of positive correlations starts only at ∼50 km, which is somewhat higher than for all other data sets. Possible reason could be that for
the SPARC climatology the SAO in the stratopause is less pronounced than for the other data sets, and is somewhat shifted in its phase. However, as in the other data sets, the upper edge of the positive correlations with $du/dz$ is at about 75 km. At around 60 km altitude the correlation between SPARC zonal wind vertical gradients and SABER gravity wave drag is somewhat weaker, possibly because this altitude range is interpolated in the SPARC climatology and might be less reliable (cf. Swinbank and Ortland, 2003).





### 7.1.3   Upper mesosphere: MSAO and effect of tides

At altitudes above about 75–80 km the positive correlation between absolute gravity wave drag and $du/dz$ does no longer hold. For the MSAO this means that the mechanisms of the gravity wave driving are somewhat different than at lower altitudes. This will be discussed in Sect. 7.2.

Above 80 km there is even an anti-correlation between absolute gravity wave drag and $du/dz$ when the SABER quasi-geostrophic winds are used also in the whole altitude range above 75 km and separated into data from ascending and descending orbit legs (Fig. 17, fifth and sixth row). The fact that this effect occurs when ascending and descending data, i.e., different local solar times, are treated separately hints at an effect of tides.

Figure 17 fifth and sixth rows, center panels, show that for ascending-only and descending-only data at altitudes above 75–80 km enhancements of absolute gravity wave drag are shifted in phase by about 180°, i.e. maxima of absolute gravity wave drag from ascending-only data fall onto minima of absolute gravity wave drag from descending-only data, and vice versa. This phase shift between the different local solar times indicates that at altitudes above 75–80 km tides have a stronger influence on the gravity wave drag distribution than the SAO. As can be seen from Figs. 17, fifth row, left, and sixth row, left, also the ascending-only and descending-only quasi-geostrophic winds are approximately in anti-phase at altitudes above about 75–80 km, which is further evidence for tidal influences.

Of course, interpolated quasi-geostrophic winds at these altitudes are affected by large biases and are not very realistic. Still, these findings show that obviously enhancements of absolute gravity wave drag are phase-locked with tidal winds and their vertical shear. This is not surprising, because it is known that gravity waves can interact with global-scale waves and can even contribute to their forcing (e.g., Holton, 1984; Smith, 2003; Matthias and Ern, 2018). Similarly, it is known from model simulations that momentum deposition of gravity waves can affect the amplitude and phase of tides (e.g., Mayr et al., 2001; England et al., 2006; Ortland and Alexander, 2006; Liu et al., 2014), and the global distribution of tides cannot be understood without gravity wave tidal interactions (e.g., Ribstein and Achatz, 2016). Also previous observations show that the distribution of gravity waves in the mesopause region is modulated by tides (e.g., Fritts and Vincent, 1987; Preusse et al., 2001; Tang et al., 2014).

However, an in-depth investigation of the impact of tides on the gravity wave distribution is beyond the scope of this paper, and for an in-depth study direction-resolved observations of momentum fluxes would be very helpful. Still, because of the anti-phase relationship between ascending and descending data, we can assume that cancellation effects will take effect if ascending and descending data are averaged, and the contribution of the SAO should become more clearly visible. This cancellation should hold for both zonal wind and absolute gravity wave drag.

## 7.2   Correlation between SABER absolute gravity wave drag and absolute zonal wind

So far we have mainly discussed the case of gravity wave forcings when a strong vertical wind shear coincides with enhancements of absolute gravity wave drag. Under these conditions, it is likely that either critical level filtering of gravity waves takes place (background winds and ground-based phase speeds become equal for parts of the gravity wave spectrum), or the vertical





gradient of the background wind leads to a reduction of intrinsic phase speeds for parts of the gravity wave spectrum such that those waves saturate and dissipate.

Of course, wave saturation can also occur independent of gradients of the background wind. If a gravity wave propagates upward conservatively in a background of constant wind and temperature, its amplitude will grow exponentially due to the decrease of atmospheric density with altitude. Upon reaching the saturation amplitude, the gravity wave will break and dissipate (e.g., Fritts, 1984). This mechanism is assumed to cause the wind reversals of the midlatitude mesospheric wind jets in the mesopause region (e.g., Lindzen, 1981). It is expected that this mechanism should also be relevant for the driving of the

MSAO, and it would explain the out-of-phase or anti-phase relationship with the SAO at lower altitudes (e.g., Dunkerton, 1982; Mengel et al., 1995), as well as the relatively slow downward propagation of the MSAO phases. While correlations between $du/dz$ and absolute gravity wave drag can be explained by critical level filtering or by gravity wave saturation, it is difficult to explain correlations between the strength of the zonal wind and absolute gravity wave drag by processes other than a general saturation mechanism of gravity waves.

First indications for a relationship between the strength of the zonal wind and absolute gravity wave drag were found in Sect. 6.3.2 for certain altitude ranges. This will now be investigated in more detail. Because only absolute values of SABER gravity wave drag are available, we will investigate in the following the correlation between *absolute values* of the zonal wind ($|u|$) and absolute gravity wave drag. A correlation analysis between absolute gravity wave drag and zonal wind, including its sign, would not make sense because correlations for situations where both positive and negative wind phases are driven

by gravity wave dissipation (as would be expected for the MSAO) would be near-zero due to cancellation effects. Correlation coefficients for the relation between $|u|$ and SABER absolute gravity wave drag are shown in the rightmost column of Fig. 17 in the same manner as before, i.e. for the different wind data sets, separately for each year, as well as for the multi-year average.

### 7.2.1   Altitudes below about 75 km

At altitudes below about 40 km the dominant mode of stratospheric variability in the tropics is the QBO. Indeed, there is some

interannual variability due to the QBO in all panels of the rightmost column of Fig. 17. However, as mentioned before, this QBO signal should be only spurious because the 10 km vertical window of our SABER momentum flux analysis will average out much of the QBO signal.

In the altitude range from about 40 to 50 km we find a positive correlation between SABER absolute gravity wave drag and $|u|$ for all wind data sets. One reason is the asymmetry of the SAO in this altitude region so that most of the positive wind

gradient falls into the negative (=westward) phase of the SAO, which is stronger than the eastward phase. Another reason is that part of the gravity wave driving, particularly during the first SAO westward phase of the year, takes place not only during eastward wind shear, but also around the line of zero wind shear, i.e. around maximum westward winds (see Fig. 17, third column). As was argued by Ern et al. (2015), this effect could be caused by gravity waves of eastward directed phase speeds that saturate before the vertical gradient $du/dz$ of the background wind becomes positive. This is supported by the fact that for

this case most of the momentum flux reduction happens at high gravity wave intrinsic phase speeds (Ern et al., 2015). Further,





during the first stratopause SAO westward phase for a given year, the gravity wave drag estimated from the reanalyses is mostly eastward (cf. Figs. 7–10). Still, as can be seen from Figs. 7–10, sometimes the net forcing can also be westward.

In the altitude range from about 50 to 75 km correlations are usually weak (floating around zero), or even negative. Strongest negative correlations are found for the SPARC zonal wind climatology (Fig. 17, lower right panel) at altitudes between 50 and
60 km, indicating that for the SPARC climatology the timing of the SAO in the lower mesosphere is somewhat different from the other data sets. Overall, the weak or negative correlations confirm that in the altitude range 50 to 75 km the SAO gravity wave driving indeed mainly happens during eastward wind shear.

### 7.2.2    Upper mesosphere and mesopause region

In the upper mesosphere, at altitudes above 75 km, the correlation between SABER absolute gravity wave drag and $du/dz$ is
usually weak, consistent with our findings in Sect. 7.1.3. Remarkably, there is a strong positive correlation between absolute gravity wave drag and $|u|$ in the altitude region of about 75 to 85 km for most of the wind data sets presented in Fig. 17. This correlation holds for each given year and also for the average year of the 2002–2018 period.

The only exception is the data set of MLS winds (see Fig. 17, seventh row, rightmost column). As MLS observations are always at two fixed local solar times, about 13:45 and 01:45 LST, tides are usually sampled at the same phase, which could
introduce altitude-dependent wind biases. Such biases could be the reason why the phases of MSAO westward winds in the altitude range 75–85 km are comparably weak, and even weaker than MSAO eastward winds. Since the strongest maxima of SABER absolute gravity wave drag are usually located in the MSAO westward phases, this potential bias will result in a weakening of correlations between SABER absolute gravity wave drag and $|u|$.

Apart from this exception, good correspondence between absolute gravity wave drag and absolute zonal wind is found for
the data sets that are based on SABER quasi-geostrophic winds merged with TIDI direct wind observations (Fig. 17, first three rows, rightmost column) and for the data sets that (in this altitude region) are based solely on SABER quasi-geostrophic winds (Fig. 17, rows four to six, rightmost column). Note that this correlation holds for ascending-only data, for descending-only data, as well as for the averages of ascending and descending data. Further, it is remarkable that the same altitude range of positive correlations is also found for the SPARC climatology (Fig. 17, bottom row, rightmost column).
At altitudes above 85 km, in the mesopause region, correlations fluctuate around zero, or are negative again. As stated in Sect. 7.1.3, this altitude region is dominated by tides, and absolute gravity wave drag seems to be phase-locked with the tidal component of the SABER quasi-geostrophic winds. An interpretation of these results, however, is difficult and beyond the scope of this study.

Overall, the positive correlations between SABER absolute gravity wave drag and absolute zonal wind speed at altitudes
75–85 km support the mechanism proposed by previous studies (e.g., Dunkerton, 1982; Mengel et al., 1995; Burrage et al., 1996) that in this altitude range selectively filtered gravity waves saturate and directly contribute to the formation of the MSAO westward and eastward phases. The direction of the wave forcing is given by the selective filtering of gravity waves at altitudes below, leading to the observed anti-correlation of the MSAO (i.e., the SAO in the upper mesosphere) and the SAO in the middle mesosphere. The wave saturation seems to take place independent of zonal wind vertical gradients, which means that gravity





waves of phase speeds much higher than the background wind are involved. The saturation amplitudes of those waves are relatively insensitive to changes in the background wind. Possibly, this explains why there is no strong downward propagation of the MSAO phases with time. For an in-depth understanding of this mechanism, however, more detailed model studies would be needed.

## 8 Correlation between reanalysis gravity wave drag and zonal wind

Next, we will investigate whether the gravity wave drag expected from the reanalyses exhibits similar characteristic patterns that are consistent with the SABER observations. Similar as in Fig. 17, Fig. 18 shows for the four reanalyses, averaged over the period 2002–2018 and latitudes 10°S–10°N: zonal wind (Fig. 18, left column), gravity wave drag overplotted with zonal wind contour lines (Fig. 18, second column), and gravity wave drag overplotted with contour lines of the zonal wind vertical gradient $du/dz$ (Fig. 18, third column). Further shown are for each altitude temporal correlations for each year, as well as for 750 the 2002–2018 average distributions between gravity wave drag and the zonal wind vertical gradient (Fig. 18, fourth column), and gravity wave drag and zonal wind including direction (Fig. 18, right column). Unlike for SABER gravity wave drag, the latter makes sense for the reanalyses because the gravity wave drag derived from the reanalyses has directionality.

### 8.1 ERA-Interim

As can be seen from Fig. 18, first row, fourth column, ERA-Interim gravity wave drag is generally positively correlated with 755 $du/dz$ with some interannual variation at altitudes below about 45 km that may be related to the QBO. This is consistent with our findings for SABER gravity wave drag (cf. Fig. 17). However, for ERA-Interim there is a strong anti-correlation between zonal wind and gravity wave drag at altitudes above ∼45 km (Fig. 18, first row, right column). This correlation is not observed for SABER gravity wave drag and should be an effect of the model sponge layer near the model top. Therefore, patterns of ERA-Interim gravity wave drag are likely not very realistic at altitudes above 45 km.

### 760 8.2 JRA-55

For JRA-55, at altitudes below ∼45 km the correlation between gravity wave drag and $du/dz$ is much stronger than for ERA-Interim, or for SABER. This indicates details in the gravity wave driving of the QBO are different in JRA-55 (cf. Fig. 18, second row, fourth column). At altitudes above ∼40 km, i.e., at altitudes even somewhat lower than for ERA-Interim, there is a strong anti-correlation between zonal wind and gravity wave drag, likely related to the model sponge layer (see Fig. 18, 765 second row, right column). Therefore, similar as for ERA-Interim, patterns of gravity wave drag are probably not very realistic at altitudes above ∼40 km.

### 8.3 MERRA-2

For MERRA-2, in the whole altitude range 30–70 km we find generally very strong positive correlation between gravity wave drag and $du/dz$ (cf. Fig. 18, third row, fourth column). Similarly to JRA-55, for altitudes below ∼45 km the MERRA-2





correlations do not show much interannual variation, which is different for ERA-Interim and SABER gravity wave drag, and
may indicate differences in details of the driving of the QBO. For the altitude range 45–70 km, the positive correlations are
qualitatively in agreement with the SABER correlations, suggesting that in this altitude range the mechanisms of the SAO
gravity wave driving may to some extent be realistic. Note that above ∼65 km the correlation between gravity wave drag and
$du/dz$ is even more positive for the years after 2004 that are fully covered by MLS observations, likely reflecting the positive
influence of assimilating MLS data in MERRA-2. At altitudes above ∼65 km, there is a strong anti-correlation between gravity
wave drag and zonal wind, which is likely caused by the sponge layer near the model top (cf. Fig. 18, third row, right column).

### 8.4 ERA-5

In the altitude range below ∼45 km ERA-5 shows interannual variability of the positive correlation between gravity wave
drag and $du/dz$ that is similar to the ERA-Interim and SABER correlations. However, in the altitude range 45–65 km there
is no clear positive correlation between gravity wave drag and $du/dz$ as would be expected from SABER observations. This
indicates that the gravity wave driving of the SSAO and the SAO in the middle mesosphere is not realistic, and might be linked
to the model imbalances that lead to the unrealistically strong eastward jets around 60 km altitude (cf. Fig. 18, fourth row,
fourth column).

The strong positive correlation in the altitude range 65–70 km seems to be related to the gravity wave drag at the top of the
eastward jets that leads to the wind reversals toward westward winds and the formation of the MSAO in ERA-5. In this altitude
range positive correlations are also found for SABER (cf. Fig. 17). However, strongest values of SABER gravity wave drag are
found at somewhat higher altitudes and are correlated with absolute wind speed rather than with $du/dz$. This correlation is not
found in ERA-5 (cf. Fig. 18, fourth row, right column). On the one hand, this means that the model sponge in ERA-5 is not as
pronounced as in the other reanalyses. On the other hand, however, it looks like not all physical mechanisms that lead to the
formation of the MSAO are correctly represented in ERA-5.

### 9 Summary and discussion

In this study, we have investigated the driving of the semiannual oscillation (SAO) of the zonal wind in the tropics by gravity
waves. The study covers the whole middle atmosphere from 30–90 km altitude and focuses on the latitude band 10°S-10°N
and the 2002–2018 time period of available satellite data.

First, the SAO was investigated in four different reanalyses, the ERA-Interim and ERA-5 reanalyses of the European Centre
for Medium-range Weather Forecasts (ECMWF), the JRA-55 reanalysis of the Japanese Meteorological Agency (JMA), and
the MERRA-2 reanalysis of the National Aeronautics and Space Agency (NASA). The expected drag due to small-scale gravity
waves was estimated as sum of the residual ("missing drag") in the transformed Eulerian mean (TEM) zonal momentum budget
and of the drag due to resolved waves of zonal wavenumbers larger than 20. All reanalyses are capable to simulate a SAO in
the stratopause region (SSAO) and show the expected asymmetry of gravity wave drag with enhanced eastward gravity wave
drag during eastward wind shear. Westward directed gravity wave drag is usually much weaker. This asymmetry is expected





because the zonal wind of the quasi-biennial oscillation (QBO) in the stratosphere has a stronger westward phase such that a larger part of the gravity wave spectrum at westward directed phase speeds encounters critical levels in the stratosphere and cannot propagate into the stratopause region and the mesosphere (cf. Dunkerton, 1982; Hamilton and Mahlmann, 1988;
Ern et al., 2015).

MERRA-2 and ERA-5 cover a larger altitude range than ERA-Interim and JRA-55. MERRA-2 applies stronger damping only above ∼58 km, uses a nonorographic gravity wave parameterization, and assimilates Microwave Limb Sounder (MLS) data in the stratosphere and mesosphere. Therefore MERRA-2 produces a reasonable SAO also in the middle mesosphere, and the SAO in the stratopause region is likely more realistic than in ERA-Interim and JRA-55. On average, also in the middle
mesosphere the eastward gravity wave driving of the SAO in MERRA-2 is stronger than the westward driving. However, there is strong interannual variability, and there are several episodes of strong westward directed gravity wave driving, for example in the year 2006. This strong inter-annual variability is also supported by satellite observations of the SAO gravity wave driving (Ern et al., 2015).

Similarly to MERRA-2, ERA-5 also uses a nonorographic gravity wave parameterization, but ERA-5 does not assimilate
MLS data. While the SSAO still looks realistic, the SAO eastward jets at altitudes around 60 km are overly strong, a fact that has already been reported in previous studies (Hersbach et al., 2018; Shepherd et al., 2018), and which was improved in the operational ECMWF model after 11 July 2017 (Hersbach et al., 2018). Among the four reanalyses investigated here, ERA-5 is the only reanalysis that simulates the mesopause SAO (MSAO) above 70 km with a strong wind reversal above the middle mesosphere SAO eastward jets.

We have also investigated the SAO based on satellite observations. According to the findings of Smith et al. (2017), quasi-geostrophic winds derived from satellite observations and interpolated into the tropics give reasonable results at altitudes below about 75–80 km. Based on quasi-geostrophic zonal winds derived from MLS observations, averaged over ascending and descending parts of the satellite orbit, we found that the SAO in the lower and middle mesosphere agrees remarkably well with the SAO in MERRA-2. Only in the upper mesosphere and in the mesopause region MLS zonal winds seem to have
an eastward bias compared to the Stratospheric Processes And their Role in Climate (SPARC) climatology of zonal winds (Swinbank and Ortland, 2003; Randel et al., 2002, 2004). Possibly, this bias is caused by tidal effects that do not completely cancel out by averaging over ascending and descending orbit data. (Data of ascending and descending MLS orbit parts are observed at different local solar time (LST), about 13:45 LST for ascending, and 01:45 LST for descending data.)

To investigate the gravity wave driving of the SAO based on satellite data, we have derived absolute gravity wave momentum
fluxes and absolute gravity wave drag from SABER temperature observations. SABER observations are not at fixed local solar times because the TIMED satellite is in a slowly precessing orbit. To capture the local solar time dependent effect of tides, as well as to account for the reduced reliability of interpolated quasi-geostrophic winds at altitudes above ∼75 km, a combined data set of ERA-Interim, interpolated SABER quasi-geostrophic winds, and winds directly observed by TIDI has been composed that should represent realistic background conditions for those gravity waves that are observed by the SABER
instrument.



We found that SABER absolute gravity wave drag has two maxima: One maximum in the stratopause region seems to be related to the SSAO, and the other maximum in the upper mesosphere to the MSAO. Further, in a large altitude range from the stratopause region, where the SSAO has its amplitude maximum, to about 75 km, SABER absolute gravity wave drag is mainly enhanced during eastward vertical wind shear $du/dz$. This modulation confirms that in the stratopause region and in

the middle mesosphere gravity waves mainly contribute to the driving of the eastward phase of the SAO and its downward propagation with time. This asymmetry is caused by the asymmetric wave filtering by the QBO in the stratosphere. Further, because slow phase speed gravity waves encounter critical levels already due to the QBO in the stratosphere, it is expected that in addition to critical level filtering also saturation of gravity waves apart from critical levels will play an important role in the stratopause region and the middle mesosphere.

In the altitude 75–80 km where the MSAO has its amplitude maximum, there is a structural change in the gravity wave interaction with the background wind. Maxima of absolute gravity wave drag are no longer observed in regions of strong $du/dz$, but in regions where the absolute zonal wind maximizes. Simultaneously, the downward propagation rate of the SAO eastward and westward wind phases is much reduced. This finding supports the theoretical expectation that gravity waves of high phase speed, that are relatively insensitive to changes in the background wind, generally saturate. Since the spectrum

is dominated by gravity waves that propagate opposite to the zonal wind in the stratopause region and middle mesosphere, this results in wave drag that is opposite to the wind direction at lower altitudes and leads to the well-known out-of-phase relationship, or even anti-correlation of the MSAO zonal wind and the SAO zonal wind at lower altitudes.

These findings were confirmed by a correlation analysis investigating the temporal correlation between SABER absolute gravity wave drag and different zonal wind data sets, separately for each year, as well as for an average over the whole period

2002–2018. It is found that the results are robust for the combined data set of SABER and TIDI winds, regardless whether ascending and descending orbit data are averaged, or whether ascending-only, or descending-only data are considered. The same is true if just SABER interpolated quasi-geostrophic winds are used, or whether winds of the SPARC climatology are used as atmospheric background. Only for the case of MLS interpolated quasi-geostrophic winds is the correlation between SABER absolute gravity wave drag and MLS absolute zonal winds in the altitude range 75–80 km widely absent, attributable

to an eastward bias of MLS winds in the upper MLT.

At altitudes above about 85 km we do not find strong correlations between SABER absolute gravity wave drag and the SAO zonal winds. Instead, absolute gravity wave drag seems to be phase-locked with the tidal wind component of the SABER interpolated quasi-geostrophic winds, which becomes most obvious if ascending and descending data are treated separately. This clearly indicates that gravity waves interact with the tides. However, an in-depth investigation of this effect is difficult and

beyond the scope of our study.

Analysis of the correlation between background wind and gravity wave drag derived from the reanalyses reveals that positive correlation between gravity wave drag and $du/dz$ is indeed found for ERA-Interim, JRA-55, and MERRA-2. ERA-Interim and JRA-55, however, are strongly limited by the sponge layers close to their model tops. Particularly, MERRA-2 seems to benefit from the assimilation of MLS data and from tuning of the gravity wave drag parameterization, such that positive

correlations between gravity wave drag and $du/dz$ are seen in a large altitude range in the mesosphere, in agreement with



SABER observations. However, MERRA-2 does not simulate a proper MSAO because it is limited by the model sponge layer above 70 km. ERA-5 does not seem to have such a strong model sponge and simulates the MSAO. However, enhanced gravity wave drag is not correlated with the magnitude of MSAO winds, which might indicate that not all parts of the gravity wave spectrum are realistically simulated by the nonorogrpahic gravity wave parameterization.

Magnitudes of SABER absolute gravity wave drag peak values are about $1$–$2\,\mathrm{m\,s^{-1}day^{-1}}$ in the stratopause region, and about $20$–$30\,\mathrm{m\,s^{-1}day^{-1}}$ in the altitude range around 80 km. These values are roughly in agreement with simulations of the SAO by free-running general circulation models (e.g., Richter and Garcia, 2006; Osprey et al., 2010; Peña–Ortiz et al., 2010), but lower than our estimates from the four reanalyses considered here, and also lower than estimates by Lieberman et al. (2010) based on TIMED observations in the mesopause region.

In a recent study, Smith et al. (2020) concluded that in free-running general circulation models, too weak gravity wave forcing would be one of the main reasons for misrepresentations of the SSAO. Indeed, we would expect that the total gravity wave driving should be stronger than indicated by the SABER absolute gravity wave drag because SABER observes only a certain part of the gravity wave spectrum (in particular, only horizontal wavelengths longer than about 100-200 km). Further, the SABER observations are affected by observational filter effects that should result in a low bias of gravity wave drag (see, for example Trinh et al., 2015; Ern et al., 2018). Still, it might be possible that SABER absolute gravity wave drag could be an overestimation, because no directional information is available, and there could be contributions of eastward and westward drag that do not cancel. However, such effects would make it difficult to explain the close relationship between positive $du/dz$ and absolute gravity wave drag in the stratopause region and middle mesosphere. Further, the SABER observations in the stratopause region are roughly in agreement with lidar observations (e.g., Deepa et al., 2006; Antonita et al., 2007) that also cover only a certain part of the whole spectrum of gravity waves. Still, because our gravity wave observations do not provide any directional information, the magnitudes of net gravity wave momentum flux and of net gravity wave drag remain an open issue that needs to be addressed by better global observations providing information about the full 3D structure of gravity waves (see also, for example Preusse et al., 2014; Ern et al., 2017; Gumbel et al., 2020).

*Data availability.*   The satellite data used in our study are open access: SABER data are available from GATS Inc. at http://saber.gats-inc.com. 895  Aura-MLS version 4.2 level 2 data are freely available via the NASA Goddard Earth Sciences Data and Information Services Center (GES DISC) at http://disc.sci.gsfc.nasa.gov/Aura. TIDI level 3 vector winds can be obtained from the National Center for Atmospheric Research (NCAR) High Altitude Observatory (HAO) website at http://timed.hao.ucar.edu/tidi/data.html.

    The ERA-5 (https://apps.ecmwf.int/data-catalogues/era5/?class=ea, last access: 1 March 2021) and ERA-Interim data (https://apps.ecmwf.int/archive-catalogue/?class=ei, last access: 1 March 2021) are available from the European Centre for Medium-Range 900  Weather Forecasts (ECMWF).

    MERRA-2 data used in this work are available at: Global Modeling and Assimilation Office (GMAO) (2015), MERRA-2 inst3_3d_asm_Nv: 3d, 3-Hourly, Instantaneous, Model-Level, Assimilation, Assimilated Meteorological Fields V5.12.4, Greenbelt, MD, USA, Goddard Earth Sciences Data and Information Services Center (GES DISC), Accessed: 1 March 2021, 10.5067/WWQSXQ8IVFW8. The MERRA-2 model level data used in this study can be accessed under https://doi.org/10.5067/QBZ6MG944HW0 (last access 1 March 2021).



JRA-55 data used in this work are available at: Japan Meteorological Agency/Japan. 2013, updated monthly. JRA-55: Japanese 55-year Reanalysis, Daily 3-Hourly and 6-Hourly Data. Research Data Archive at the National Center for Atmospheric Research, Computational and Information Systems Laboratory. https://doi.org/10.5065/D6HH6H41. Accessed 1 March 2021.

The SPARC temperature and zonal wind climatology is available at:

http://www.sparc-climate.org/data-center/data-access/reference-climatologies/randels-climatologies/temperature-wind-climatology/. Accessed
1 March 2021.

*Author contributions.*  ME designed and performed the technical analysis. MD contributed to the processing and analysis of the reanalysis data. MD, PP, and MR contributed with ideas to the scientific interpretation of results. MGM, MJS, and QW provided important information about the instruments and their data used in this study. All coauthors contributed to the interpretation of results and preparation of the paper.

*Competing interests.*  The authors declare that they have no conflict of interests.

*Acknowledgements.*  We would like to thank the teams of the MLS, SABER and TIDI instruments, as well as the SPARC data center for creating and maintaining the excellent data sets used in our study. The authors are grateful to the European Centre for Medium-Range Weather Forecasts (ECMWF), the Japan Meteorological Agency (JMS), and NASA's Global Modeling and Assimilation Office (GMAO) for providing the reanalysis data used in this work, as well as the Research Data Archive at the National Center for Atmospheric Research (NCAR) for distribution of the JRA-55 reanalysis.

*Financial support.*  This work was supported by the Federal German Ministry for Education and Research (Bundesministerium für Bildung und Forschung, BMBF) project QUBICC, grant number 01LG1905C, which is part of the Role of the Middle Atmosphere in Climate II (ROMIC-II) program of BMBF. This work was also supported by the Deutsche Forschungsgemeinschaft (DFG, German Research Foundation) projects PR 919/4–2 and ER 474/4–2 (MS–GWaves/SV) which are part of the DFG research unit FOR 1898 (MS–GWaves). The work by MD was supported by DFG project DI 2618/1–1. Work at the Jet Propulsion Laboratory, California Institute of Technology, was
done under contract with NASA (80NM0018D0004). TIDI operations are supported by NASA Grant NNX17AG69G. MGM acknowledges ongoing support from the NASA Heliophysics Division in support of the SABER instrument.





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





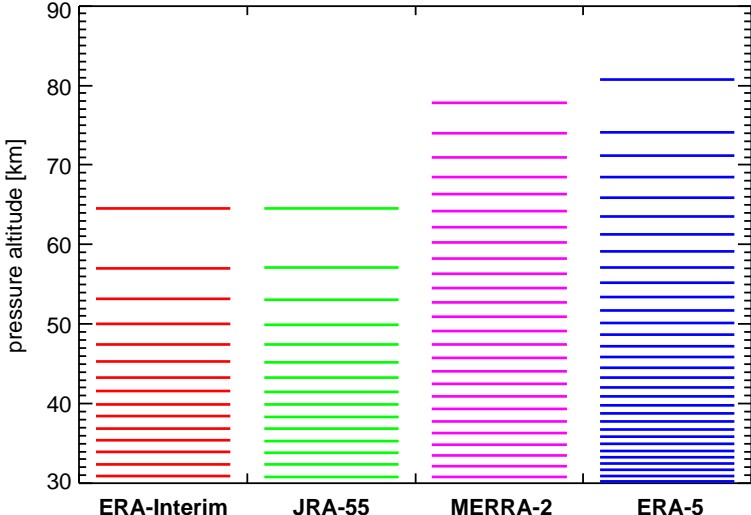

**Figure 1.** Altitude levels of the four reanalyses in the approximate altitude range 30 to 90 km used in this study. Altitudes given in this figure are pressure altitudes using a fixed pressure scale height of 7 km.



**Figure 2.** Local solar times of SABER and TIDI observations at the equator for the 2002–2018 time period. Local solar times are given for the following: SABER ascending (black diamonds), SABER descending (black crosses) TIDI warm side ascending (red diamonds), TIDI warm side descending (red crosses), TIDI cold side ascending (blue diamonds), and TIDI cold side descending (blue crosses).

**Figure 3.** ERA-Interim zonal-average zonal wind averaged over $10°$S–$10°$N for the 2002–2018 time period together with the multi-year mean seasonal cycle over the same period (lower row, middle panel). For comparison the lower right panel shows the corresponding zonal winds of the SPARC climatology (cf. Swinbank and Ortland, 2003; Randel et al., 2002, 2004). Contour line increment is $20\,\mathrm{m\,s^{-1}}$. The zero wind line is highlighted in bold solid, and westward (eastward) winds are indicated by dashed (solid) contour lines.

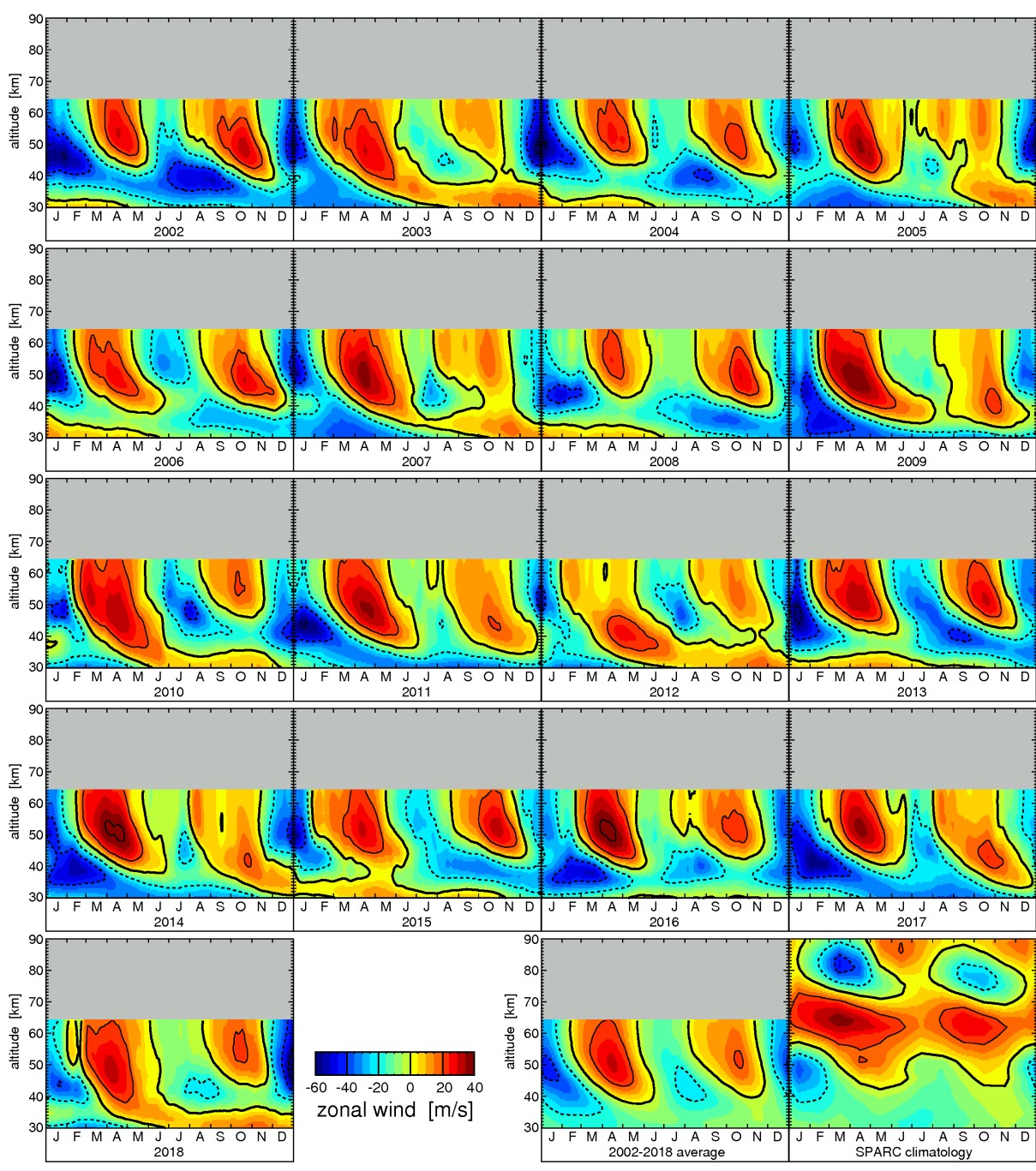

**Figure 4.** Same as Fig. 3, but for the JRA-55 reanalysis.

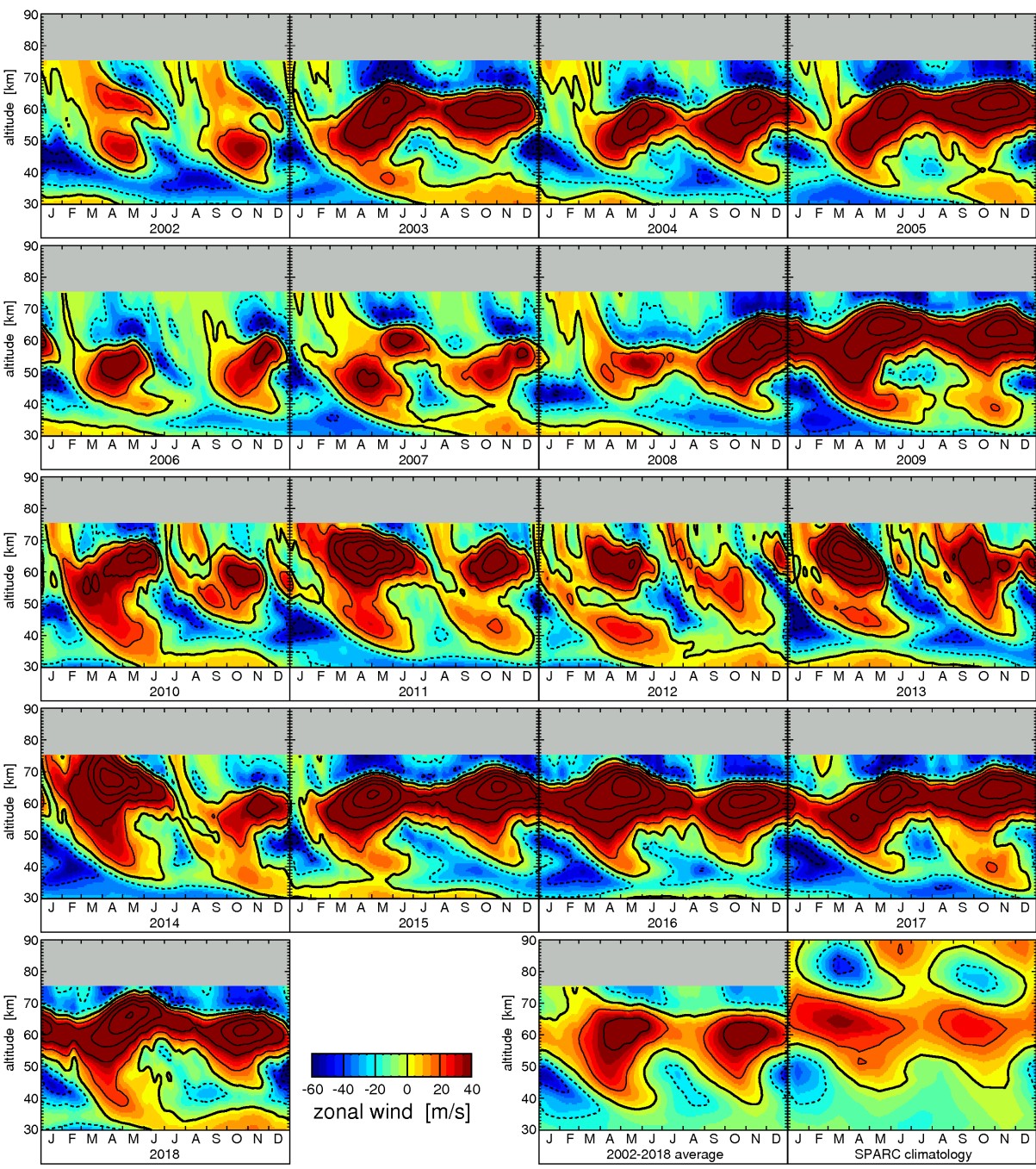

**Figure 5.** Same as Fig. 3, but for the ERA-5 reanalysis.

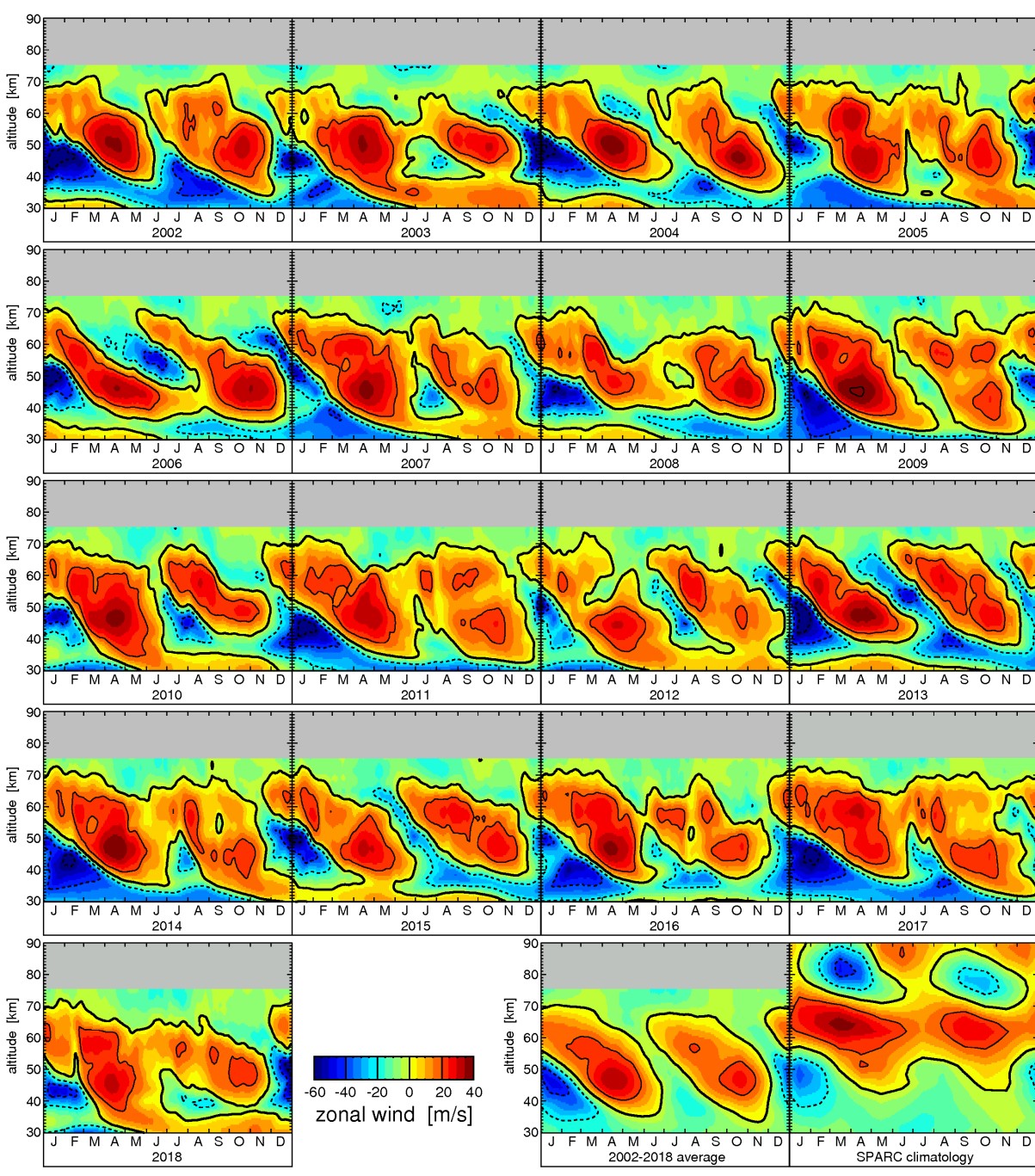

**Figure 6.** Same as Fig. 3, but for MERRA-2.



**Figure 7.** ERA-Interim estimates of gravity wave drag averaged over $10°S–10°N$ for the 2002–2018 time period, as well as the multi-year average over these years. Overplotted contour lines are ERA-Interim zonal winds averaged over $10°S–10°N$ (cf. Fig 3. Contour line increment is $20\,\mathrm{m\,s^{-1}}$. The zero line is highlighted in bold solid, and westward (eastward) zonal wind is indicated by dashed (solid) contour lines.

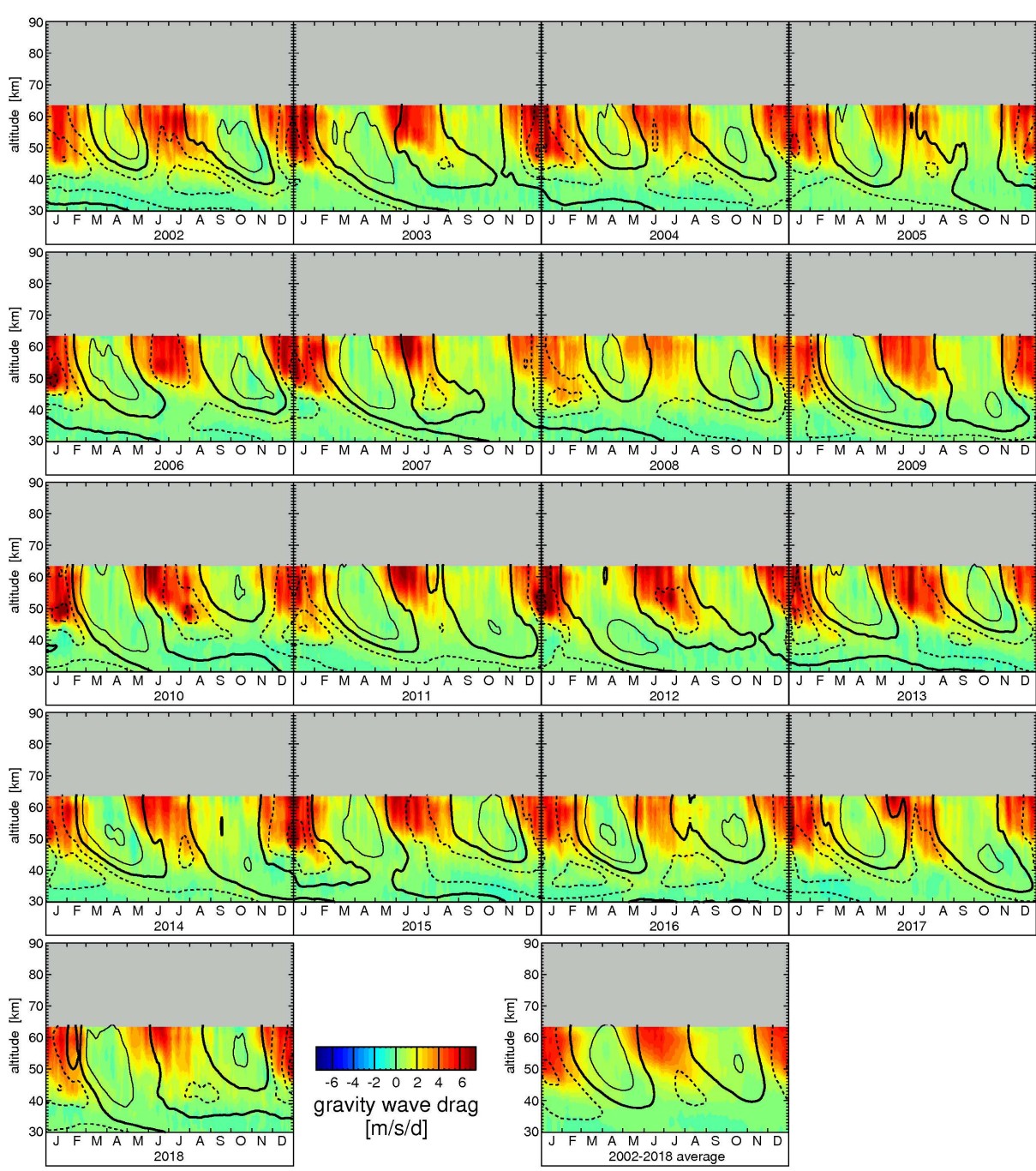

**Figure 8.** Same as Fig. 7, but for JRA-55.

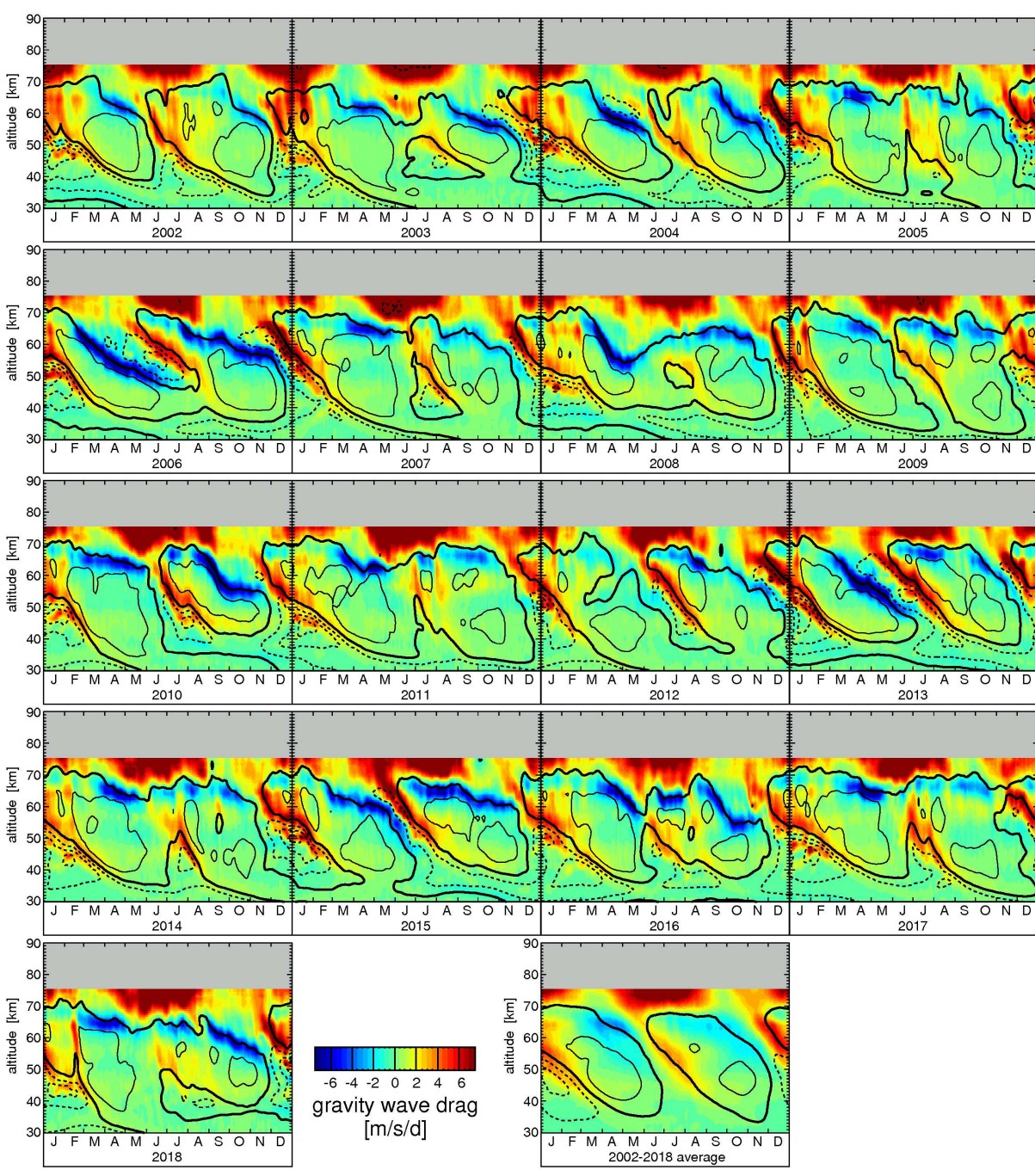

**Figure 9.** Same as Fig. 7, but for MERRA-2.





**Figure 10.** Same as Fig. 7, but for ERA-5.



**Figure 11.** Same as Fig. 3, but for interpolated MLS quasi-geostrophic zonal winds averaged over the latitude band 10°S–10°N. These winds represent an average over both local solar times (ascending and descending orbit nodes are combined). MLS observations started in 2004. Therefore the panels for 2002 and 2003 are left blank.



**Figure 12.** Same as Fig. 3, but for the combined data set of ERA-Interim zonal winds below 35 km, interpolated SABER quasi-geostrophic zonal winds at altitudes 45–75 km, and a smooth transition between ERA-Interim and SABER winds at altitudes 35–45 km. Winds above ∼80 km are TIDI cold side zonal winds. Several smaller TIDI data gaps are closed by linear interpolation in time. For one larger data gap and in the beginning of 2002 SABER quasi-geostrophic winds are used also above 75 km. Otherwise the data gap between SABER winds at 75 km and TIDI winds at 80 km is interpolated. Winds are an average over 10°S–10°N, and SABER and TIDI winds are an average over ascending and descending orbit branches.

**Figure 13.** SABER absolute gravity wave momentum flux in mPa on a logarithmic scale for the 2002–2018 time period averaged over the latitude band 10°S–10°N, and the average over these years (lower right panel). Contour lines are zonal-average zonal winds of the merged data set based on ERA-Interim, SABER, and TIDI, as described in Sect. 5.3.

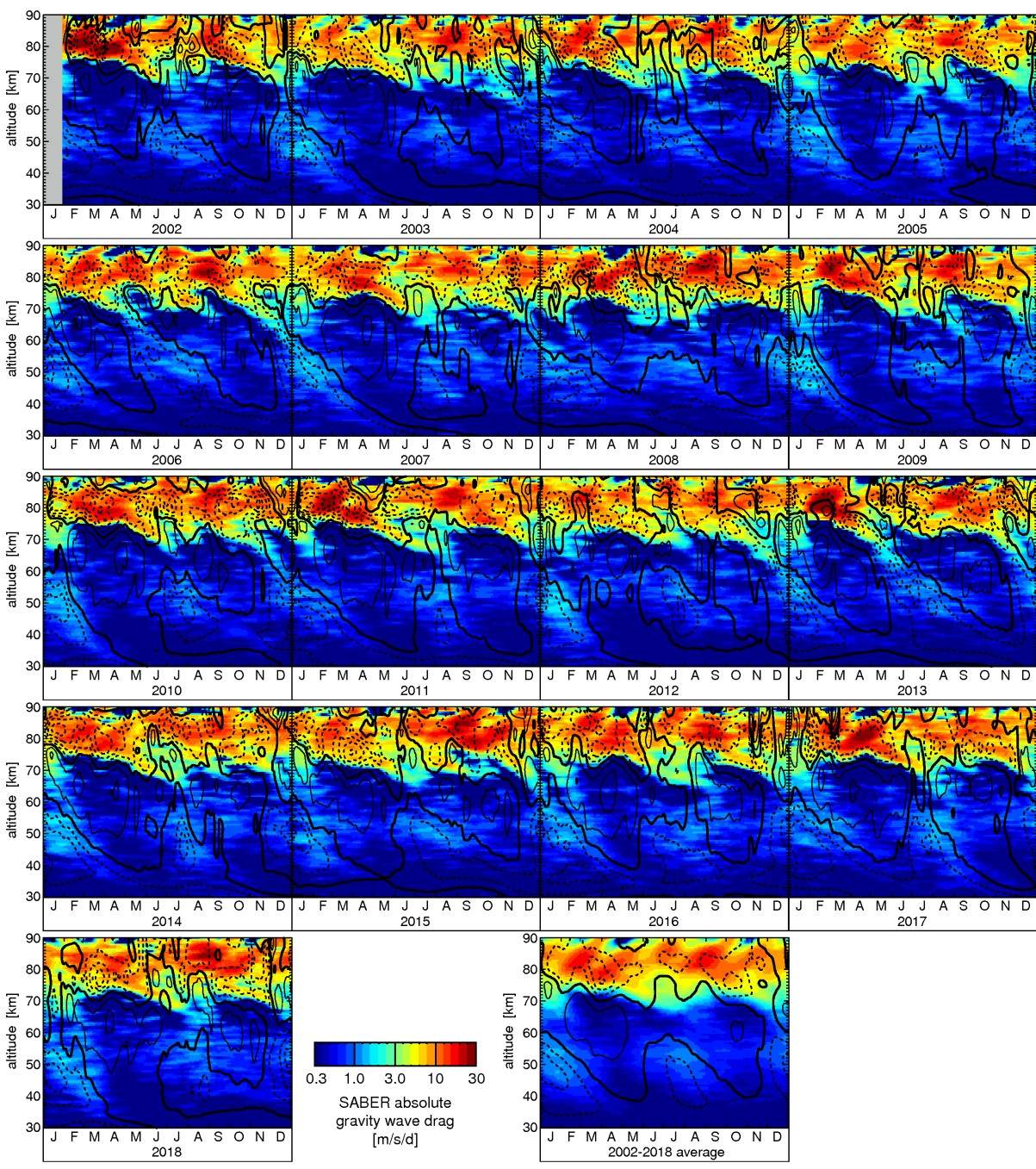

**Figure 14.** SABER absolute gravity wave drag in $\mathrm{m\,s^{-1}day^{-1}}$ on a logarithmic scale for the 2002–2018 time period averaged over the latitude band $10°S–10°N$, and the average over these years (lower right panel). Contour lines are zonal-average zonal winds of the merged data set based on ERA-Interim, SABER, and TIDI, as described in Sect. 5.3.

**Figure 15.** Same as Fig. 14, but for the gravity wave drag anomaly, i.e., the absolute gravity wave drag shown in Fig. 14 normalized by the average gravity wave drag, separately for each year and altitude.



**Figure 16.** Same as Fig. 15, but contour lines are the vertical gradient of zonal-average zonal winds of the merged data set based on ERA-Interim, SABER, and TIDI, as described in Sect. 5.3. Contour lines are at 0, $\pm 2$, $\pm 5$, and $\pm 10\,\mathrm{m\,s^{-1}\,km^{-1}}$. Westward (=negative) gradients are indicated by dashed contour lines.





**Figure 17.** Left column: Different zonal wind data sets averaged over $10°$ S–$10°$ N and the years 2002–2018 (2004–2018 for MLS). Second column: SABER absolute gravity wave drag averaged over the same latitudes and corresponding periods, overplotted with zonal wind contour lines. Third column: Same as second column, but for normalized SABER absolute gravity wave drag overplotted with contour lines of zonal wind vertical gradients. Fourth column: Temporal correlations between SABER gravity wave drag and zonal wind vertical gradients, separately for each year, and for the multi-year averages. Right column: Same as fourth column, but for the correlation between SABER gravity wave drag and zonal wind absolute values. The different rows are (from top to bottom) for (1) the merged ERA-Interim, SABER and TIDI data set as described in Sect. 5.3 averaged over ascending and descending orbit legs, (2) same as (1), but only for ascending orbit legs, (3) same as (1), but only for descending orbit legs, (4) ERA-Interim and SABER winds merged, similar as for MLS in Sect. 5.2, averaged over ascending and descending orbit legs (i.e. SABER geostrophic winds are used also above 75 km), (5) same as (4), but only for ascending orbit legs, (6) same as (4), but only for descending orbit legs, (7) ERA-Interim and MLS winds merged as described in Sect. 5.2, averaged over both orbit legs, and (8) SPARC climatology winds (same wind used for each year).



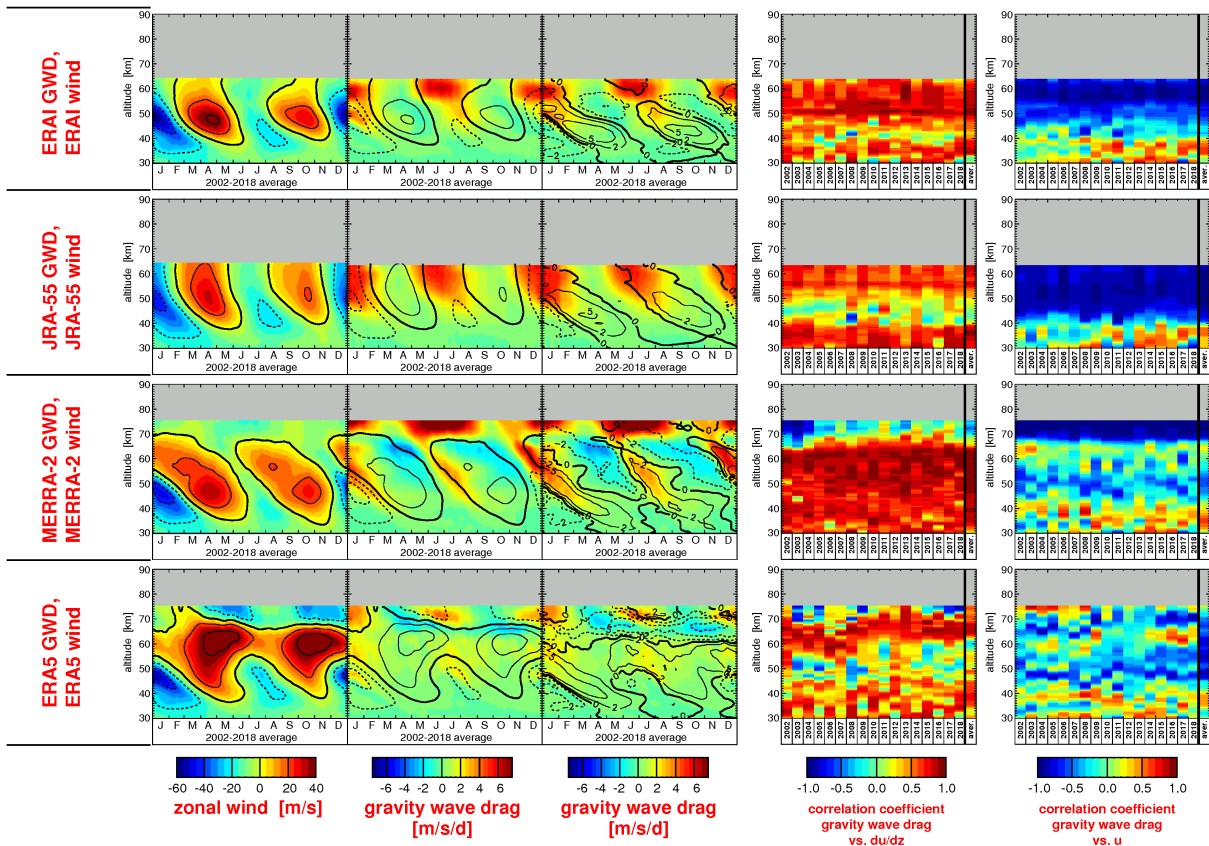

**Figure 18.** Left column: zonal winds of the four reanalyses considered in this study, averaged over the latitude band $10°$S–$10°$N and the years 2002–2018. Second column: Gravity wave drag in $\mathrm{m\,s^{-1}day^{-1}}$ derived from the different reanalyses and averaged over the years 2002 until 2018, overplotted with zonal wind contour lines. Third column: Same as second column, but contour lines are the vertical gradient of the zonal wind. Fourth column: Temporal correlations between reanalysis gravity wave drag and zonal wind vertical gradients, separately for each year, and for the averages over the different years. Right column: Temporal correlations between reanalysis gravity wave drag and zonal wind, separately for each year, and for the averages over the different years. The different rows are (from top to bottom) for (1) ERA-Interim, (2) JRA-55, (3) MERRA-2, and (4) ERA-5.