# Peer review of "The semiannual oscillation (SAO) in the tropical middle atmosphere and its gravity wave driving in reanalyses and satellite observations"

_Atmospheric Chemistry and Physics, 2021_

## Author Comment (AC1)

Many thanks to Referee #1 for appreciating our work and the very helpful comments that will significantly help to improve the manuscript!

Please find below our point-by-point reply to the reviewer concerns. Comments by Reviewer #1 are given in red, our reply is given in black, and changes in the manuscript are indicated in blue.

**Reply to the Main Concerns by Reviewer # 1:**

**(Main Concern 1a:) Only the mean annual cycle of the different variables should be shown and should be combined into different figures, according to parameters. The time evolutions could be included as supplemental material.**

As recommended, we will move the single-year figures into the supplemental material, and we will merge figures in the main paper.

We will also remove the column showing zonal winds in former Figs. 17 and 18 (the correlation figures) because these figures were quite crowded, too, and the column showing zonal winds is helpful for illustration, but redundant. Further, from former Fig. 17 we will move three of the rows into the Supplement.

**(Main Concern 1b) Sections should be rearranged to discuss the SAO first, and the GW driving thereafter.**

The sections in the paper will be rearranged, as recommended.

**(Main Concern 2) Given the differences between the SAO signal in reanalyses and the SPARC climatology, it seems difficult to believe the "total" GW drag estimations. Therefore, the mean annual cycle of the different terms of the GW drag calculations (i.e. resolved, parameterized and the residual) should be shown to better understand what the different terms are are doing in each reanalysis.**

Of course, it is not expected that the reanalyses simulate a "perfect" SAO. However, there are features that are common in all datasets: for example, the first SAO period of a year is stronger. Because our knowledge of the SAO and its driving is very poor, estimates of the gravity wave (GW) driving of the SAO from reanalyses will provide important information about the mechanisms that drive the SAO. For this information only relative variations of the GW driving are needed, and not the exact magnitude that might not be very robust.
This will be pointed out more clearly at the beginning of Sect. 5 in the revised manuscript.

Indeed, in the later sections of the paper it turns out that in all reanalyses the GW driving is prevalently eastward in the lower mesosphere, which is in agreement with the satellite observations.
(It should also be kept in mind that, as already mentioned in the paper, the SAO in MERRA-2 is in relatively good agreement with the wind products derived from satellite observations and may therefore be more reliable.)

As explained in the main paper, the total zonal GW drag $\overline{X}_{GW}$ in models consists of three different contributions and can be written as follows:

$$\overline{X}_{GW} = \overline{X}_{res}(k > 20) + \overline{X}_{param} + \overline{X}_{imbalance} \qquad (1)$$

with $\overline{X}_{res}(k > 20)$ the GW drag due to model-resolved waves with zonal wavenumbers $k > 20$, $\overline{X}_{param}$ the parameterized zonal GW drag, and $\overline{X}_{imbalance}$ the "residual", or remaining imbalance that is caused, for example, by data assimilation.

As recommended, in a new Sect. 5.2, we will include and discuss in the revised manuscript also the resolved GW drag $\overline{X}_{res}(k > 20)$ for all reanalyses, and the parameterized GW drag

$\overline{X}_{param}$ for JRA-55 and MERRA-2. For a further illustration and discussion, the remaining imbalance $\overline{X}_{imbalance}$ can then be calculated for JRA-55 and MERRA-2 as the remaining difference.

For ERA-Interim and ERA-5, parameterized GW drag is not available from the ECMWF MARS archive!

Please note that in the recent paper by Gupta et al. (2021) one of the ERA-5 momentum terms in their paper is named "parameterized drag term (PGWD)". However, as becomes obvious from their Eq. (1), **their term "PGWD" is not the parameterized GW drag** $\overline{X}_{param}$, but the sum $\overline{X}_{param} + \overline{X}_{imbalance}$, i.e., the sum of parameterized GW drag and the residual (=remaining imbalance)!

**In the following, we will introduce and discuss the different contributions to the "total" GW drag that will be added in the revised paper:**

Figure 1 of this reply shows the total gravity wave drag (GWD) $\overline{X}_{GW}$ for the four reanalyses, and Fig. 2 the drag $\overline{X}_{res}(k > 20)$ due to resolved waves of $k > 20$. Figure 3 shows the parameterized GW drag $\overline{X}_{param}$ for JRA-55 and for MERRA-2, and Fig. 4 the remaining imbalance $\overline{X}_{imbalance}$ for JRA-55 and for MERRA-2. For Figs. 1–4 all values are averages over the period 2002–2018 and the latitude band 10S–10N.

[Figure]

**Figure 1:** Total zonal GW drag $\overline{X}_{GW}$ for **(a)** ERA-Interim, **(b)** JRA-55, **(c)** ERA-5, and **(d)** MERRA-2. Overplotted are contour lines of the corresponding zonal average zonal winds for the respective reanalysis dataset. Contour line interval is 20 m/s. The zero wind line is highlighted in bold solid, and westward (eastward) winds are indicated by dashed (solid) contour lines.

As can be seen from Figs. 1 and 2 of this reply, the resolved GW drag is negligible in ERA-Interim, JRA-55, and MERRA-2. (Please note that in Fig. 1 the range of the color scale is $\pm 7.5$ m/s/d, while it is only $\pm 0.25$ m/s/d in Fig. 2a, 2b, and 2d, and only $\pm 1.25$ m/s/d in Fig. 2c.) Only for ERA-5 below 55 km $\overline{X}_{res}(k > 20)$ sometimes contributes as much as about 50% to

[Figure]

**Figure 2:** Resolved GW drag $\overline{X}_{res}(k > 20)$ for **(a)** ERA-Interim, **(b)** JRA-55, **(c)** ERA-5, and **(d)** MERRA-2. Overplotted are contour lines of the corresponding zonal average zonal winds for the respective reanalysis dataset. Contour line interval is 20 m/s. The zero wind line is highlighted in bold solid, and westward (eastward) winds are indicated by dashed (solid) contour lines.

$\overline{X}_{GW}$. In both $\overline{X}_{GW}$ and $\overline{X}_{res}(k > 20)$ eastward GW drag is stronger than westward GW drag in the upper stratosphere and lower mesosphere, which is a consequence of the QBO wave filtering in the stratosphere below.

As can be seen from Fig. 3a, the parameterized GW drag $\overline{X}_{param}$ is closely linked with the background wind and opposite to it. This is expected because JRA-55 does not have an explicit nonorographic GW parameterization and uses only Rayleigh friction at upper levels. A similar distribution would be expected for ERA-Interim, because, similar as JRA-55, ERA-Interim uses Rayleigh friction at upper levels and does not have a nonorographic GW parameterization.

Comparing Fig. 1d and Fig. 3b, it is evident that for MERRA-2 in the whole altitude range $\overline{X}_{GW}$ and $\overline{X}_{param}$ are almost the same.

As can be seen from Fig. 4a, for JRA-55, above 40 km the remaining imbalance is strongly positive. This likely indicates that a really large positive assimilation increment is needed to compensate the unrealistic effect of Rayleigh friction, and to keep the model temperature and winds in agreement with assimilated observations. The situation should be similar for ERA-Interim.

For MERRA-2, $\overline{X}_{imbalance}$ (Fig. 4b) is close to zero. Apparently, in the tropics, the nonorographic GW drag scheme of MERRA-2 has been tuned in a way to minimize the assimilation increment caused by the assimilation of MLS and other data (see also Molod et al., 2005).

[Figure]

**Figure 3:** Parameterized GW drag $\overline{X}_{param}$ for **(a)** JRA-55, and **(b)** MERRA-2. Please note that from the GES-DISC archive MERRA-2 parameterized GW drag is not available for the whole altitude range. Overplotted are contour lines of the corresponding zonal average zonal winds for the respective reanalysis dataset. Contour line interval is 20 m/s. The zero wind line is highlighted in bold solid, and westward (eastward) winds are indicated by dashed (solid) contour lines.

[Figure]

**Figure 4:** Remaining imbalance $\overline{X}_{imbalance}$ for **(a)** JRA-55, and **(b)** MERRA-2. Overplotted are contour lines of the corresponding zonal average zonal winds for the respective reanalysis dataset. Contour line interval is 20 m/s. The zero wind line is highlighted in bold solid, and westward (eastward) winds are indicated by dashed (solid) contour lines.

This should be the reason why MERRA-2 simulates a reasonable SAO even in the years when MLS data were still not available (in the period prior to August 2004).

Overall, our results show that there are differences between the different reanalyses that reflect the different stages of model development. In particular, our results demonstrate that the use of a nonorographic gravity wave parameterization can be very useful because it can be tuned in a way to produce more realistic results (as was seen for MERRA-2). Even though there are strong differences in the model setups, there are similarities of the total gravity wave drag, particularly in the stratopause region (total GW drag is predominately eastward).

**References:**

Gupta, A., Birner, T., Dornbrack, A., and Polichtchouk, I.: Importance of gravity wave forcing for springtime southern polar vortex breakdown as revealed by ERA5, Geophysical Research Letters, 48, e2021GL092762, https://doi.org/10.1029/2021GL092762, 2021.

Molod, A., Takacs, L., Suarez, M., and Bacmeister, J.: Development of the GEOS-5 atmospheric general circulation model: evolution from MERRA to MERRA2, Geosci. Model Dev., 8, 1339-1356, doi:10.5194/gmd-8-1339-2015, 2015.

**(Main Concern 3a)** **Limitations of the momentum flux calculations from SABER should be discussed in more detail, in particular the effect of the trajectory of the satellite being perpendicular to the average GW wavenumber vector.**

As recommended, we will add more discussion about the shortcoming of using SABER along-track GW horizontal wavenumbers, instead of "true" GW horizontal wavenumbers.

Generally, the use of along-track GW horizontal wavenumbers as a proxy for the true GW horizontal wavenumbers will lead to a low-bias of SABER momentum fluxes (the momentum flux is proportional to the horizontal wavenumber) because the along-track GW horizontal wavenumber will always underestimate the true horizontal wavenumber.

The AIRS satellite instrument has a similar orbit geometry. Because AIRS provides 3D temperature observations, it is possible to determine from AIRS observations true GW horizontal wavenumbers, as well as along-track GW horizontal wavenumbers. This opportunity has been taken by Ern et al. (2017) to compare true and along-track GW horizontal wavenumbers: AIRS observations indicate an underestimation of the along-track wavenumber (corresponding to an underestimation of momentum fluxes) by a factor between 1.5 and somewhat above 2.

In addition, for SABER there will be aliasing effects (undersampling of observed GWs) and effects of the instrument sensitivity function of limb sounding satellite instruments (cf. Ern et al., 2018), which should both lead to an even stronger underestimation of GW momentum fluxes. Therefore the error of SABER GW momentum fluxes should be at least a factor of two, and momentum fluxes are likely strongly underestimated.

This detailed discussion will be included in the revised manuscript in the newly introduced Sect. 6.1.1.

**References:**

Ern, M., Hoffmann, L., and Preusse, P.: Directional gravity wave momentum fluxes in the stratosphere derived from high-resolution AIRS temperature data, Geophys. Res. Lett., 44, 475-485, doi:10.1002/2016GL072007, 2017.

Ern, M., Trinh, Q. T., Preusse, P., Gille, J. C., Mlynczak, M. G., Russell III, J. M., and Riese, M.: GRACILE: A comprehensive climatology of atmospheric gravity wave parameters based on satellite limb soundings, Earth Syst. Sci. Data, 10, 857-892, doi:10.5194/essd-10-857-2018, 2018.

**(Main Concern 3b)** **The expression SABER "absolute GW drag" should be avoided. For example, the expression "vertical derivative of the absolute momentum flux" could be used, instead.**

The reviewer is correct that only under certain conditions the vertical gradient of GW momentum flux can be used to calculate a proxy for absolute net gravity wave drag. However, if these conditions are met, convincing results were obtained in a number of previous studies. This is why, for convenience, we used the expression "absolute gravity wave drag" in the paper.

The problem is that the absolute GW drag proxy is not simply the vertical gradient of absolute GW momentum flux, but it is also normalized by $-1/\varrho$, with $\varrho$ the background density. This makes it difficult to find a short expression for the absolute GW drag proxy.

Therefore, in order to avoid the impression that the vertical gradient of GW momentum flux would always give reliable results, we will introduce an abbreviation which will make the reader check how this proxy is introduced, and what are its limitations:

"SABER MFz-proxy-|GWD|"

To make sure that this introduction is not easily overread, former Sect. 6.1 will be split into two subsections, Sect. 6.1.1 addressing SABER absolute momentum fluxes and its limitations (see **Main Concern 3a**), and Sect. 6.1.2 addressing the SABER GW drag proxy. In Sect. 6.1.2 we will also add more discussion on the limitations of the SABER GW drag proxy.

**Reply to the Minor Comments by Reviewer # 1:**

**(Minor Comment 1) lines 578-585. Could there be an effect of GWs with zonal momentum flux being very few at these high altitudes due to critical level filtering by the QBO and the SSAO?**

Thank you very much for this comment!
Of course, GW filtering by the QBO and SSAO will strongly reduce GW momentum fluxes in the tropical mesopause region. This may be one of the reasons why the mesopause SAO (MSAO) occurs only in a narrow layer.

This will be mentioned in the revised paper in Sect. 6.3.3.

**(Minor Comment 2) lines 595. Normalized SABER GW drag is the same as GW drag anomalies in Fig. 14?**

In order to avoid confusion, the expression "anomaly" will no longer be used in the revised paper.

**(Minor Comment 3) Fig. 17, line 598. Why not showing the correlation over the whole period of study, instead of the correlation of the multiyear mean annual cycle? What would be the difference between the two, and its interpretation?**

Given the strong interannual variability of the SAO, it is most important that the correlation holds for the majority of single years. This is why the single-year correlations are shown in our paper. The average year is just shown to illustrate that the correlation even holds for the average year — even if the average year might be affected by strong outlier-years, or compensation effects. Correlations over the whole period could also be affected by strong outliers, but in a different way than the average year.
For completeness, we will show the correlation over the whole dataset as an additional column. It turns out that these additional columns give results which are in most cases very similar to those of the average years, or an average-by-eye over the single years.

**(Minor Comment 4) Lines 734-743. What is the explanation of a high correlation between the so-called (SABER) absolute GW drag (see main comment #3) and the zonal wind speed, if saturation of GWs due to decrease in density is the proposed mechanism?**

There are at least two situations when it makes sense to use SABER MFz-proxy-|GWD| as a proxy for absolute net GW drag:

(1) The GW spectrum is dominated by slow and moderate phase speeds opposite to the background wind:
A layer of (initial) wind shear reducing the wind speed will also reduce the intrinsic phase speed of the GWs dominating the spectrum and bring them closer to saturation. This will lead to a stronger dissipation of those waves and thereby lead to a strengthening and downward propagation of the wind shear layers and wind bands. This effect is seen for the QBO and its wave-driven downward propagation of eastward and westward wind bands. In the middle and lower mesosphere, this effect is also seen for the SAO and its downward propagating eastward wind bands, but only less pronounced for the westward wind bands because GWs of westward phase speeds are weaker due to wind filtering by the asymmetric QBO winds in the stratosphere.
For this mechanism, one would expect correlation between SABER MFz-proxy-|GWD| and zonal wind absolute vertical gradients.

(2) The GW spectrum is dominated by fast phase speed GWs with a directional preference
Minor variations of the background wind will have only little effect on the GW phase speeds

and thus the GW saturation amplitudes. Due to the decrease of atmospheric density with altitude, GW amplitudes will grow exponentially to conserve momentum flux while propagating upward. At some point, however, these GWs will saturate and exert drag on the background flow. As these GWs are not much influenced by the background wind and its variation, the saturation altitude (the altitude where the waves exert their drag) will not be as closely tied to a wind shear zone as for situation (1), but can lead to a reversal and strengthening of the wind by inducing a *temporal wind tendency* (and not to a strengthening of the vertical wind shear and eventually to a downward propagation of the shear zone). The temporal wind tendency will lead to a wind reversal and wind strengthening at the same altitude where the drag is exerted.

Therefore, enhanced GW drag should be observed at the same altitude as the reversed wind jet and lead to a correlation between SABER MFz-proxy-|GWD| and (absolute) wind speed. Situation (2) should match the conditions for the region of the MSAO around 80km altitude and may explain why there is no strong downward propagation of MSAO eastward and westward wind phases.

Case (1) is already discussed in-depth in the paper. However, the reviewer is correct that the discussion of Case (2) was still not very clear. Therefore, we will include the above discussion of Case (2) at the end of Sect. 7.2.2 in the revised manuscript.

**(Minor Comment 5) In the same spirit, how can ERA-5 have a realistic GW driving of the SAO if the SAO in ERA-5 is not realistic (section 8.4)?**

We do not think that the GW driving of the SAO in ERA-5 is fully realistic:
As already mentioned in Sect. 8.4, in ERA-5 the SAO and its gravity wave driving in the upper stratosphere to middle mesosphere is not considered to be realistic, because there are overly strong eastward wind jets.
At even higher altitudes, the nonorographic GW drag scheme in ERA5 seems to be able to induce a mesopause SAO (MSAO). Even for this altitude range, we find that the characteristics of ERA5 GW drag related to the MSAO show differences to the SABER observations. The are two possible reasons for this: First, the poor representation of the SAO in the lower and middle mesosphere in ERA5 will lead to an incorrect filtering of the GW spectrum, and thus to a not fully realistic forcing of the MSAO. Second, the GW spectrum may be incorrect already at the source level. This means that also the forcing of the MSAO in ERA5 might not be really physically sound.

Obviously, the reviewer comment addresses the question whether the ERA5 gravity wave driving of the MSAO could still be realistic.
This will be addressed at the end of Sect. 8.4 by stating more clearly that differences to SABER observations hint at an underrepresentation of high phase speed GWs in ERA-5, i.e., not all physical mechanisms that lead to the formation of the MSAO are correctly represented in ERA-5. In addition, we mention that the unrealistic SAO at lower altitudes can lead to an unrealistic wind filtering of the gravity wave spectrum, which can also affect the simulation of the MSAO.

**(Minor Comment 6) Lines 771-773. I do not understand what the authors mean here. Since MERRA-2 assimilates MLS observations, the driving of the SAO in MERRA-2 at 45-70km is likely the result of this process. But this does not mean that the GW driving of the SAO in MERRA-2 is realistic.**

As is indicated in Figs. 1–4 of this reply, the nonorographic GW drag scheme in MERRA-2 was tuned in a way to minimize the assimilation increment due to MLS observations. This means that the SAO in MERRA-2 is mainly a result of the tuned nonorographic GW drag scheme, and might therefore be better also because of more realistic GW drag. Please note that the SAO in MERRA-2 is generally reasonable — even before 2004 when no MLS data are available and the model is relatively unconstrained in the middle mesosphere.

Further, the qualitative agreement between the SABER and MERRA-2 correlations seems to indicate that — at least to some extent — the physical mechanisms of the GW driving of the SAO are realistically simulated by the MERRA-2 nonorographic GW drag scheme.

This additional discussion will be included in the revised paper in Sect. 8.3.

In addition, we will address the four technical comments — thank you very much for finding these inaccuracies!

---

## Author Comment (AC2)

We would like to thank Referee #2 for appreciating our efforts and for the very helpful comments and suggestions that will definitively improve the manuscript!

Please find below our point-by-point reply to the reviewer concerns. Comments by Reviewer #2 are given in red, our reply is given in black, and changes in the manuscript are indicated in blue.

**Reply to the Main Concerns by Reviewer # 2:**

**(Main Concern 1a:) l.270-272 It is unclear whether a purely zonal approach of assuming k>20 is adequate to extract model-resolved GWs. This would only be the case if zonal propagation of GWs is dominant in the tropics. Otherwise, a more sophisticated approach as proposed by Watanabe et al., 2008, or Becker et al., 2018 would be needed. The contribution of resolved GWs should be shown for more information.**

Of course, some care has to be taken whether the major part of the model-inherent gravity wave (GW) drag is from model-resolved GWs, and whether methods as suggested by Reviewer 2 should be applied.

As suggested, we will include the following two references in the revised manuscript as a guidance for readers who want to analyze model-resolved GWs in detail:

Watanabe, S., Kawatani, Y., Tomikawa, Y., Miyazaki, K., Takahashi, M., and Sato, K.: General aspects of a T213L256 middle atmosphere general circulation model, J. Geophys. Res., 113, D12110, doi:10.1029/2008JD010026, 2008.

Becker, E., and Vadas, S. L.: Secondary gravity waves in the winter mesosphere: Results from a high-resolution global circulation model, J. Geophys. Res.: Atmospheres, 123, 2605-2627, https://doi.org/10.1002/2017JD027460, 2018.

However, there are three reasons why we think that a more sophisticated approach is not required in our paper:

- The GW drag of model-resolved waves with k>21 is only a minor part of the GW drag proxy derived in our paper. This is the case because the spatial resolution of the reanalyses is relatively coarse. As a consequence, they resolve only a limited part of the GW spectrum. This can be seen below from Figs. 1 and 2 which show our estimates of the total and the resolved GW drag. Uncertainties in the resolved GW drag will therefore not much affect our estimates of total GW drag. Introducing a limit of k>21 is somewhat arbitrary, anyhow.

- In the tropics, zonal propagation of GWs should be dominant because zonal winds are usually stronger than meridional winds, and GWs propagating opposite to the background wind can attain larger amplitudes because the intrinsic phase speed of these waves, and thus their saturation amplitudes, are increased. Therefore, the purely zonal analysis should be justified even for the part of the wave spectrum that is resolved in the models (in the tropics, many other studies also use this kind of zonal approach — for example, even the KANTO model group did so in the follow-up paper Kawatani et al., 2010, with S. Watanabe as one of the coauthors).

- The validity of a zonal-only approach is also supported by the fact that at most altitudes all our GW drag proxies (reanalyses and SABER) show strong correlation with either the zonal wind, or its vertical gradient.

These three points will be mentioned in the revised manuscript in the new Sect. 5.2.1, In addition, we will include in the paper — among other new figures — the figure showing the resolved GW drag for the four reanalyses.

**A brief discussion of Figs. 1 and 2:**

As explained in the main paper, the total zonal GW drag $\overline{X}_{GW}$ in models consists of three different contributions and can be written as follows:

$$\overline{X}_{GW} = \overline{X}_{res}(k > 20) + \overline{X}_{param} + \overline{X}_{imbalance} \tag{1}$$

with $\overline{X}_{res}(k > 20)$ the GW drag due to model-resolved waves with zonal wavenumbers $k > 20$, $\overline{X}_{param}$ the parameterized zonal GW drag, and $\overline{X}_{imbalance}$ the "residual", or remaining imbalance that is caused, for example, by data assimilation.

[Figure]

**Figure 1:** Total zonal GW drag $\overline{X}_{GW}$ for **(a)** ERA-Interim, **(b)** JRA-55, **(c)** ERA-5, and **(d)** MERRA-2. Overplotted are contour lines of the corresponding zonal average zonal winds for the respective reanalysis dataset. Contour line interval is 20 m/s. The zero wind line is highlighted in bold solid, and westward (eastward) winds are indicated by dashed (solid) contour lines.

As can be seen from Figs. 1 and 2 of this reply, the resolved GW drag is negligible in ERA-Interim, JRA-55, and MERRA-2. (Please note that in Fig. 1 the range of the color scale is $\pm 7.5$ m/s/d, while it is only $\pm 0.25$ m/s/d in Fig. 2a, 2b, and 2d, and only $\pm 1.25$ m/s/d in Fig. 2c.) Only for ERA-5 below 55 km $\overline{X}_{res}(k > 20)$ sometimes contributes as much as about 50% to $\overline{X}_{GW}$. In both $\overline{X}_{GW}$ and $\overline{X}_{res}(k > 20)$ eastward GW drag is stronger than westward GW drag in the upper stratosphere and lower mesosphere, which is a consequence of the QBO wave filtering in the stratosphere below.

**References:**

Kawatani, Y., Sato, K., Dunkerton, T. J., Watanabe, S., Miyahara, S., and Takahashi, M.: The roles of equatorial trapped waves and internal inertia gravity waves in driving the quasi-biennial oscillation. Part I: Zonal mean wave forcing, J. Atmos. Sci., 67, 963-980, doi:10.1175/2009JAS3222.1, 2010.

[Figure]

**Figure 2:** Resolved GW drag $\overline{X}_{res}(k > 20)$ for **(a)** ERA-Interim, **(b)** JRA-55, **(c)** ERA-5, and **(d)** MERRA-2. Overplotted are contour lines of the corresponding zonal average zonal winds for the respective reanalysis dataset. Contour line interval is 20 m/s. The zero wind line is highlighted in bold solid, and westward (eastward) winds are indicated by dashed (solid) contour lines.

**(Main Concern 1b)** Assuming zonal propagation of GWs may contradict the flux estimate of SABER, in which the along-track wavelength is regarded as the horizontal wavelength. If zonal propagation of GWs is dominant in the tropics, an error of flux estimate could be large. Please mention the error of flux estimate in more detail.

This concern is the same as **Main Concern 3a** by Reviewer 1. Therefore our corresponding reply is repeated here:

As recommended, We will add more discussion about the shortcoming of using SABER along-track GW horizontal wavenumbers, instead of "true" GW horizontal wavenumbers.

Generally, the use of along-track GW horizontal wavenumbers as a proxy for the true GW horizontal wavenumbers will lead to a low-bias of SABER momentum fluxes (the momentum flux is proportional to the horizontal wavenumber) because the along-track GW horizontal wavenumber will always underestimate the true horizontal wavenumber.
The AIRS satellite instrument has a similar orbit geometry. Because AIRS provides 3D temperature observations, it is possible to determine from AIRS observations true GW horizontal wavenumbers, as well as along-track GW horizontal wavenumbers. This opportunity has been taken by Ern et al. (2017) to compare true and along-track GW horizontal wavenumbers: AIRS observations indicate an underestimation of the along-track wavenumber (corresponding to an underestimation of momentum fluxes) by a factor between 1.5 and somewhat above

2.

In addition, for SABER there will be aliasing effects (undersampling of observed GWs) and effects of the instrument sensitivity function of limb sounding satellite instruments (cf. Ern et al., 2018), which should both lead to an even stronger underestimation of GW momentum fluxes. Therefore the error of SABER GW momentum fluxes should be at least a factor of two, and momentum fluxes are likely strongly underestimated.

This detailed discussion will be included in the revised manuscript in the newly introduced Sect. 6.1.1.

**References:**

Ern, M., Hoffmann, L., and Preusse, P.: Directional gravity wave momentum fluxes in the stratosphere derived from high-resolution AIRS temperature data, Geophys. Res. Lett., 44, 475-485, doi:10.1002/2016GL072007, 2017.

Ern, M., Trinh, Q. T., Preusse, P., Gille, J. C., Mlynczak, M. G., Russell III, J. M., and Riese, M.: GRACILE: A comprehensive climatology of atmospheric gravity wave parameters based on satellite limb soundings, Earth Syst. Sci. Data, 10, 857-892, doi:10.5194/essd-10-857-2018, 2018.

**(Main Concern 2) Figures for respective years are shown throughout this paper. However, there is little discussion of interannual variation. It is better to reduce unnecessary figures and make them larger and easier to see.**

This comment is similar to **Main Concern 1a** by Reviewer 1.

We will move the single-year figures into the supplemental material and only keep the multi-year averages in the main paper. Further, we will merge figures in the main paper.
In addition, in former Figs. 17 and 18 we will omit the column showing the zonal wind, because it is redundant. Further, we will move three rows of former Fig. 17 into the Supplement.

**(Main Concern 3) It is mentioned that the effect of tide is large for satellite data above 80 km and contaminate the GW contribution to the SAO driving throughout this paper. Although I understand that it is the signature of the interaction between the tide and GWs and important, it looks far from the primary purpose of this paper. I recommend moving it to another paper.**

Because the local time of SABER observations is continuously changing, a brief discussion of local-time effects, including tides, cannot be avoided. Further, the correlation between SABER vertical gradients of absolute momentum flux and the background winds has very different characteristics in different altitude ranges.
For a full understanding, these differences need to be explained, even if these differences are caused by the QBO or tides, and not directly by the SAO. Otherwise, readers will start to question our methods.

Therefore we still mention the effect of tides, but significantly shorten the discussion — particularly in Sects. 7.1.3 and 7.2.2.

**Reply to the Minor Comments by Reviewer # 2:**

**(Minor Comment 1) The SPARC climatology is regarded "true" and the difference from it is expressed as "bias" throughout this paper. However, it may be better to change the expression because it could be caused by the difference in time resolution and interannual variation as shown in L. 225-228. At least, it is better to compare monthly averages between SPARC climatology and reanalysis/satellite data, which clarify the effect of time resolution.**

As recommended, the word "bias" has been removed in connection with the SPARC climatology.

Just to clarify: we do not regard the SPARC climatology as "true"!
We show the SPARC climatology just for comparison because our general knowledge of the SAO is very poor. In spite of its shortcomings, this climatology gives some information on the basic structure of the SAO, which is helpful to guide the discussion throughout the paper. This might have caused the impression we would consider the SPARC climatology to be the "truth".

In order to clearly emphasize that the SPARC climatology is not the "truth", we feel that more discussion on the weaknesses of this climatology is needed. A number of weaknesses of the URAP/SPARC climatology is listed below, and a brief version of this listing will be included as additional discussion in the revised paper in Sect. 4.1.

**Some potential limitations of the URAP/SPARC climatology:**

- The SPARC climatology is based on the URAP zonal wind climatology, which uses direct wind observations by the HRDI instrument. HRDI observes the Doppler shift of spectral lines from satellite. An inherent problem of this method is that the zero-wind is not clearly defined (e.g., Hays et al., 1993; Baron et al., ACP, 2013), which involves assumptions and may introduce biases.

- HRDI uses only daytime data and covers only 50 to 75% of local times in the mesosphere (cf. Fig. 2 in Swinbank et al. 2003) which can introduce biases, although a correction of tidal effects was attempted.

- URAP uses HRDI observations from the 7 years 1992-1998, but HRDI temporal coverage is reduced to much less than ∼50% after mid 1996. Strictly speaking, this means that the SPARC climatology contains only a short period of 4.5 years of quasi-continuous HRDI observations. Only this period should be more reliable because directly guided by HRDI observations. Consequently, interannual variability will still have strong effect on the monthly averages of the SPARC climatology.

- Spatial and temporal gaps in the HRDI wind observations are filled by a climatology, or by interpolation based on model data (UKMO analyses) and balanced winds (cf. Randel et al., 2002). This combination of different data sets, as well as the data processing, may introduce certain biases. Multi-year averaging over a longer dataset with good temporal resolution (like in our paper) was not possible for the SPARC climatology due to the shortness and further shortcomings of the datasets.

- There is a HRDI data gap (no observations) centered around 0.3 hPa ( 55km, cf. Swinbank 2003, their p.2, and their Figs. 3a and 6) that needs to be filled by climatology, model data, or interpolation, and could introduce biases. This makes the continuously eastward directed winds at 60km in SPARC questionable. Particularly, because there is strong interannual variability, including also periods of **westward** directed winds, at that

altitude in all our datasets. Therefore, already in the ACPD paper version we dedicated some discussion to the question whether this feature in the URAP/SPARC climatology could be reliable.

Given these facts, it should be clear that the SPARC climatology can contain biases and should not considered to be the "truth".

Since the SPARC climatology is not considered to be a reference, or the "truth", it does not make much sense to, for example, reduce the time resolution of our better resolved datasets to that of the SPARC climatology. Further, we hope that the better time resolution of our dataset may be considered helpful by others.

**References:**

Baron, P., Murtagh, D. P., Urban, J., Sagawa, H., Ochiai, S., Kasai, Y., Kikuchi, K., Khosrawi, K., Körnich, H., Mizobuchi, S., Sagi, K., and Yasui, M.: Observation of horizontal winds in the middle-atmosphere between 30°S and 55°N during the northern winter 2009–2010, Atmos. Chem. Phys., 13, 6049–6064, doi:10.5194/acp-13-6049-2013, 2013.

Hays, P. B., Arbreu, V. J., Dobbs, M. E., Gell, D. A., Grassl, H. J., and Skinner, W. R.: The high-resolution Doppler imager on the Upper Atmosphere Research Satellite, J. Geophys. Res., 98, 10713–10723, 1993.

Randel, W., Chanin, M.–L., Michaut, C., and the SPARC Reference Climatology Group: SPARC intercomparison of middle atmosphere climatologies, WCRP 116, WMO/TD No. 1142, SPARC report No. 3, 2002.

Swinbank, R., and Ortland, D. A.: Compilation of wind data for the UARS Reference Atmosphere Project, J. Geophys. Res., 108, 4615, doi:10.1029/2002JD003135, 2003.

**(Minor Comment 2) It seems unnecessary to discuss the correlation in QBO and sponge layers in this paper.**

The Rayleigh drag exerted in the sponge layers is one of the contributions of the GW drag proxy. For JRA-55 and ERA-Interim, this is even one of the main contributions. In the discussion in Sect. 8 we point out that this contribution is not considered to be very realistic. We feel that this discussion is needed because Reviewer 1 in his **Major Concern 2** was worried that the residual drag from reanalyses would not always be representative of GW drag. For this reason, we keep this discussion in order not to appear to be too uncritical about our results. In addition, the altitude where the negative correlation between the residual drag and the zonal wind strengthens gives valuable information above which altitude the residual drag becomes increasingly unrealistic.

Therefore we decided to keep the discussion about Rayleigh drag in the sponge layers.

The biennial variations seen in our correlation analysis are a striking feature that needs to be explained. Without an explanation, readers would think this correlation would be an artifact, and doubt our methods. Overall, the discussion of the QBO-correlations does not take much space, anyhow.

Therefore, for completeness, we decided to keep the discussion related to the QBO.

---

## Author Response (AR1)

Dear Editor, Dear Reviewers,

We would like to thank the Editor and both Reviewers for their effort and their very constructive comments!

Please find enclosed in this response file:

- the point-by-point reply to the Reviewer Comments
- a marked-up manuscript version showing the changes made
- the new supplement that contains a number of single-year plots and parts of one of the former correlation plots

Main changes in the revised manuscript are:

- · All figures based on single years have been moved into a supplement
- Sections have been rearranged to first discuss the SAO, and only later its driving by gravity waves
- The different contributions to the total gravity wave drag of the reanalyses are shown and discussed
- We have added more discussion on limitations of absolute momentum fluxes and the proxy for absolute gravity wave drag derived from satellite data

We hope that in the revised manuscript all concerns were adequately addressed.

With best regards, Manfred Ern

**Reply to Reviewer 1**

Many thanks to Referee #1 for appreciating our work and the very helpful comments that will significantly help to improve the manuscript!

Please find below our point-by-point reply to the reviewer concerns. Comments by Reviewer #1 are given in red, our reply is given in black, and changes in the manuscript are indicated in blue.

**Reply to the Main Concerns by Reviewer # 1:**

(Main Concern 1a:) Only the mean annual cycle of the different variables should be shown and should be combined into different figures, according to parameters. The time evolutions could be included as supplemental material.

As recommended, we will move the single-year figures into the supplemental material, and we will merge figures in the main paper.

We will also remove the column showing zonal winds in former Figs. 17 and 18 (the correlation figures) because these figures were quite crowded, too, and the column showing zonal winds is helpful for illustration, but redundant. Further, from former Fig. 17 we will move three of the rows into the Supplement.

**(Main Concern 1b) Sections should be rearranged to discuss the SAO first, and the GW driving thereafter.**

The sections in the paper will be rearranged, as recommended.

(Main Concern 2) Given the differences between the SAO signal in reanalyses and the SPARC climatology, it seems difficult to believe the "total" GW drag estimations. Therefore, the mean annual cycle of the different terms of the GW drag calculations (i.e. resolved, parameterized and the residual) should be shown to better understand what the different terms are are doing in each reanalysis.

Of course, it is not expected that the reanalyses simulate a "perfect" SAO. However, there are features that are common in all datasets: for example, the first SAO period of a year is stronger. Because our knowledge of the SAO and its driving is very poor, estimates of the gravity wave (GW) driving of the SAO from reanalyses will provide important information about the mechanisms that drive the SAO. For this information only relative variations of the GW driving are needed, and not the exact magnitude that might not be very robust.

This will be pointed out more clearly at the beginning of Sect. 5 in the revised manuscript.

Indeed, in the later sections of the paper it turns out that in all reanalyses the GW driving is prevalently eastward in the lower mesosphere, which is in agreement with the satellite observations.

(It should also be kept in mind that, as already mentioned in the paper, the SAO in MERRA-2 is in relatively good agreement with the wind products derived from satellite observations and may therefore be more reliable.)

As explained in the main paper, the total zonal GW drag  $\overline{X}_{GW}$  in models consists of three different contributions and can be written as follows:

$$\overline{X}_{GW} = \overline{X}_{res}(k > 20) + \overline{X}_{param} + \overline{X}_{imbalance}$$
(1)

with  $\overline{X}_{res}(k > 20)$  the GW drag due to model-resolved waves with zonal wavenumbers k > 20,  $\overline{X}_{param}$  the parameterized zonal GW drag, and  $\overline{X}_{imbalance}$  the "residual", or remaining imbalance that is caused, for example, by data assimilation.

As recommended, in a new Sect. 5.2, we will include and discuss in the revised manuscript also the resolved GW drag  $\overline{X}_{res}(k > 20)$  for all reanalyses, and the parameterized GW drag  $\overline{X}_{param}$  for JRA-55 and MERRA-2. For a further illustration and discussion, the remaining imbalance  $\overline{X}_{imbalance}$  can then be calculated for JRA-55 and MERRA-2 as the remaining difference.

For ERA-Interim and ERA-5, parameterized GW drag is not available from the ECMWF MARS archive!

Please note that in the recent paper by Gupta et al. (2021) one of the ERA-5 momentum terms in their paper is named "parameterized drag term (PGWD)". However, as becomes obvious from their Eq. (1), their term "PGWD" is not the parameterized GW drag  $\overline{X}_{param}$ , but the sum  $\overline{X}_{param} + \overline{X}_{imbalance}$ , i.e., the sum of parameterized GW drag and the residual (=remaining imbalance)!

**In the following, we will introduce and discuss the different contributions to the "total" GW drag that will be added in the revised paper:**

Figure 1 of this reply shows the total gravity wave drag (GWD)  $\overline{X}_{GW}$  for the four reanalyses, and Fig. 2 the drag  $\overline{X}_{res}(k > 20)$  due to resolved waves of k > 20. Figure 3 shows the parameterized GW drag  $\overline{X}_{param}$  for JRA-55 and for MERRA-2, and Fig. 4 the remaining imbalance  $\overline{X}_{imbalance}$  for JRA-55 and for MERRA-2. For Figs. 1–4 all values are averages over the period 2002–2018 and the latitude band 10S–10N.

**Figure 1:** Total zonal GW drag  $\overline{X}_{GW}$  for (a) ERA-Interim, (b) JRA-55, (c) ERA-5, and (d) MERRA-2. Overplotted are contour lines of the corresponding zonal average zonal winds for the respective reanalysis dataset. Contour line interval is 20 m/s. The zero wind line is highlighted in bold solid, and westward (eastward) winds are indicated by dashed (solid) contour lines.

As can be seen from Figs. 1 and 2 of this reply, the resolved GW drag is negligible in ERA-Interim, JRA-55, and MERRA-2. (Please note that in Fig. 1 the range of the color scale is